# READING IMAGES LIKE TEXTS: SEQUENTIAL IMAGE UNDERSTANDING IN VISION-LANGUAGE MODELS

**Yueyan Li, Chenggong Zhao, Zeyuan Zang, Caixia Yuan,** * **Xiaojie Wang**
Beijing University of Posts and Telecommunications
{siriuslala,zcggoon,zangzeyuan,yuancx,xjwang}@bupt.edu.cn

## ABSTRACT

Vision-Language Models (VLMs) have demonstrated remarkable performance across a variety of real-world tasks. However, existing VLMs typically process visual information by serializing images, a method that diverges significantly from the parallel nature of human vision. Moreover, their opaque internal mechanisms hinder both deeper understanding and architectural innovation. Inspired by the dual-stream hypothesis of human vision, which distinguishes the "what" and "where" pathways, we deconstruct the visual processing in VLMs into object recognition and spatial perception for separate study. For object recognition, we convert images into text token maps and find that the model's perception of image content unfolds as a two-stage process from shallow to deep layers, beginning with attribute recognition and culminating in semantic disambiguation. For spatial perception, we theoretically derive and empirically verify the geometric structure underlying the positional representation in VLMs. Based on these findings, we introduce an instruction-agnostic token compression algorithm based on a plug-and-play visual decoder to improve decoding efficiency, and a RoPE scaling technique to enhance spatial reasoning. Through rigorous experiments, our work validates these analyses, offering a deeper understanding of VLM internals and providing clear principles for designing more capable future architectures .

## 1 INTRODUCTION

Generative vision-language models (VLMs) take images and texts as input and generate texts as output. Existing VLMs are typically based on the Transformer architecture (Vaswani et al., 2017) and have achieved strong performance on numerous real-world tasks. However, VLMs are prone to significant hallucination issues, such as incorrectly describing objects or misjudging the spatial relationships between them. These problems not only present challenges for model improvement but also highlight the need for interpretability research, motivating a deeper investigation into the underlying mechanisms behind a model to foster a more comprehensive understanding of it.

The existing research on the interpretability of multimodal models is very limited. Some studies focus on the internal representations of a model, finding the neurons or components (e.g., attention heads) corresponding to the concepts in the real world (Goh et al., 2021; Gandelsman et al., 2023; 2024), or extracting semantic information using logit lens (Neo et al., 2024; Sonia Joseph, 2024; Jiang et al., 2024) or sparse autoencoders (Hugofry, 2024; Daujotas, 2024; Rao et al., 2024). Another line of work employs causal tracing to investigate information storage and transfer (Rajaram et al., 2024; Palit et al., 2023; Basu et al., 2024) or uses blocking-based interventions to study the information flow (Zhang et al., 2024b; 2025) in VLMs. However, current research often overlooks a fundamental distinction between neural networks and human cognition. Specifically, the visual encoder in a VLM is typically a Vision Transformer (ViT) (Dosovitskiy et al., 2020) that adopts the Transformer architecture originally designed for inherently sequential text. When processing an image, a ViT employs *raster scan*: it partitions the image into patches and flattens them into a one-dimensional sequence. The adjacent patches belonging to the same object may be scattered in different and discontinuous positions in that sequence. In contrast, human visual perception exhibits

---

*Corresponding author: yuancx@bupt.edu.cn
[1]Code is available at https://github.com/Siriuslala/vlm_interp.

characteristics of *Gestalt cognition* (Koffka, 2013), where the brain actively organizes and integrates discontinuous visual signals upon reception, viewing them as a whole. This discrepancy raises critical questions: How do VLMs leverage a 1D image sequence to understand complex 2D concepts and perform tasks like spatial reasoning? And does this cognitive gap between machine and human processing adversely affect VLM performance?

The dual-stream hypothesis (Goodale & Milner, 1992) of the human brain posits that visual processing is divided into a ventral pathway for object recognition (the "what pathway") and a dorsal pathway for spatial perception (the "where pathway"). Inspired by this theory, our work focuses on these two aspects and poses two primary questions: (1) *How do VLMs associate positionally discontinuous tokens belonging to the same object within a flattened sequence to predict the object's category?* (2) *How do VLMs infer 2D spatial relationships between objects from a 1D sequence?*

To address question (1), we conduct an investigation into the visual information processing within the VLM's visual encoder, combining quantitative and visualization methods. Specifically, we focus on the encoder's representations layer by layer, employing analyses based on logit lens. We find that the visual encoder exhibits a two-stage process from shallow to deep layers: it first performs attribute recognition, identifying local features such as color and texture, and then leverages the attention mechanism for semantic disambiguation to assign these features to a specific object category. This two-stage process collectively resembles a pattern of Gestalt cognition. For question (2), we begin by analyzing the properties of learnable 1D absolute position embeddings, such as those in LLaVA (Liu et al., 2023b). We then focus on 2D Rotary Position Embeddings (RoPE) (Su et al., 2024), the most common method for achieving dynamic resolution. We theoretically analyze how a 2D RoPE-based visual encoder represents positional information, and empirically visualize the geometric properties of the resulting positional representation, confirming the validity of our theoretical analysis.

To further validate the efficacy and practical value of the aforementioned findings, we propose two corresponding model improvements. First, based on our discoveries in object recognition, we introduce a novel instruction-agnostic token compression algorithm. We first distill a visual decoder that takes visual embeddings as input and outputs the logits for visual tokens, and the latter can be mapped to text tokens. Then we compress the visual embeddings at the pre-filling stage using run-length encoding to merge the similar visual tokens, making it more efficient for inference. Second, informed by our analysis of spatial perception, we propose the RoPE scaling algorithm to address the issue of indistinct positional representation in RoPE-based visual encoders. This method enhances the model's spatial reasoning capabilities by adaptively amplifying positional information in low-frequency regions. RoPE scaling demonstrates strong performance in both training-free and fine-tuning-based experiments. In summary, our main contributions are as follows:

- For the "what" pathway in a VLM, we present an in-depth analysis of the dynamic visual information processing in the VLM's visual encoder based on logit lens and visualization, uncovering a two-stage processing pattern analogous to Gestalt cognition (Section 3).
- For the "where" pathway in a VLM, we provide a theoretical analysis of the mechanism behind spatial perception for RoPE-based visual encoders, and reveal the geometric structure of the representation of spatial relationships through empirical studies (Section 4).
- Based on the visual processing characteristics of VLMs, we propose an instruction-agnostic token compression method that reduces image sequence length during the decoding phase while limiting performance loss to an acceptable range (Section 5.1).
- Based on the spatial perception characteristics of VLMs, we introduce RoPE scaling, a method that improves the spatial reasoning capabilities of VLMs based on 2D RoPE while preserving their general capabilities (Section 5.2).

## 2 BACKGROUND

### 2.1 VLM ARCHITECTURE

As shown in Figure 1, a VLM typically consists of a visual encoder $f_I(\cdot)$, a modality connector $f_C(\cdot)$, and a language model $f_T(\cdot)$. Suppose the input to a VLM is an image $X^V$ and an instruction $X^T$. The instruction is tokenized and mapped to a sequence of text embeddings $T = (t_1, ..., t_{N_T}) \in \mathbb{R}^{N_T \times D}$, where $N_T$ is the number of text tokens and $D$ is the dimension of the language model.

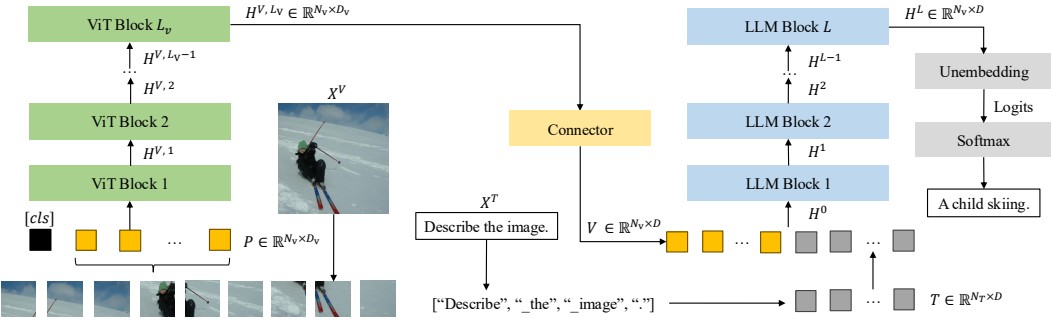

Figure 1: The typical architecture of the vision-language model. It consists of an visual encoder, a modality connector and a decoder-only language model. Notations are detailed in Section 2.1

For the image input, the visual encoder is a ViT with bidirectional attention. First, the image is partitioned into a sequence of patches $P = (p_1, ..., p_{N_V}) \in \mathbb{R}^{N_V \times D_V}$, where $N_V$ is the number of patches and $D_V$ is the dimension of the visual encoder. Then the sequence goes through $L_V$ layers of ViT blocks, and the output of layer $l(l \in [1, L_V])$ is $H^{V,l} = (x_1^l, ..., x_{N_V}^l) \in \mathbb{R}^{N_V \times D_V}$. The modality connector is a projection (e.g. MLP). It maps the output of the visual encoder $H_V^{L_V}$ to the semantic space of texts, which results in the visual embeddings $V = (v_1, ..., v_{N_V}) \in \mathbb{R}^{N_V \times D}$. Finally, the embeddings of the image and the instruction are concatenated as a multimodal input $H^0 = (v_1, ..., v_{N_V}, t_1, ..., t_{N_T}) \in \mathbb{R}^{(N_V + N_T) \times D}$ of the language model for response generation.

## 2.2 POSITION EMBEDDING

Position embedding is introduced to add positional information to input, and thus the computation of the attention weights in Transformer is position-aware. It can be categorized into absolute position embedding and relative position embedding. The absolute position embedding is directly added to the input embedding before being passed into the Transformer blocks. Let $X = (x_1, ..., x_n) \in \mathbb{R}^{n \times d}$ be a $d$-dimensional input embedding with $E = (e_1, ..., e_n) \in \mathbb{R}^{n \times d}$ being its absolute position embedding. Then the input with absolute position embedding is:

$$X + E = (x_1 + e_1, ..., x_n + e_n) \in \mathbb{R}^{n \times d} \tag{1}$$

In practice, the absolute position embedding could be either pre-calculated by a sinusoidal function (Vaswani et al., 2017) or a set of learnable vectors (Radford et al., 2018; Devlin et al., 2019). Instead of assigning a unique embedding to each position, relative position embedding focuses on the relative distances between different positions. The most commonly used one is RoPE (Su et al., 2024), which encodes the absolute position with a rotation matrix and incorporates the explicit relative position dependency in self-attention formulation in each layer. Considering a query $q_m = W_Q x_m = q_m^{(0)} + i q_m^{(1)}$ and a key $k_n = W_K x_n = k_n^{(0)} + i k_n^{(1)} (m \neq n)$ in a 2-dimensional input $X \in \mathbb{R}^{n \times 2}$ at two different positions $m$ and $n$, RoPE applies a transformation $f$ to them and the inner product between the query and the key is calculated as follows:

$$< f(q_m, m), f(k_n, n) > = Re[q_m e^{im\theta} \cdot (k_n e^{in\theta})^*] = Re[q_m k_n^* e^{i(m-n)\theta}] \tag{2}$$

where $\theta$ is a preset constant. In equation 2, the relative distance $m - n$ is introduced during the calculation of the attention weights. For 2D RoPE in VLMs, the position IDs corresponding to the width and the height of a patch are both introduced in RoPE. Given two positions $(m_1, m_2)$ and $(n_1, n_2)$ for the query and the key, the query is at least 4D and can be written as $q_m = (q_m^X, q_m^Y) = (q_m^{(0)} + i q_m^{(1)}, q_m^{(2)} + i q_m^{(3)})$, where $X$ and $Y$ denote the components along the width and the height directions, respectively. The same applies to the key as well. Then the inner product is:

$$\begin{aligned} < f(q_m, m_1, m_2), f(k_n, n_1, n_2) > &= Re\big[ (q_m^X e^{im_1\theta}, q_m^Y e^{im_2\theta}) \cdot ((k_n^X e^{in_1\theta})^*, (k_n^Y e^{in_2\theta})^*) \big] \\ &= Re\big[ q_m^X k_n^{X*} e^{i(m_1-n_1)\theta} + q_m^Y k_n^{Y*} e^{i(m_2-n_2)\theta} \big] \end{aligned} \tag{3}$$

In this work, the VLMs we use include LLaVA-1.5 (Liu et al., 2024a), Qwen2 / 2.5-VL (Wang et al., 2024; Bai et al., 2025) and InternVL-2.5 (Chen et al., 2024b). The Qwen2 / 2.5-VL series of models use 2D RoPE in the visual encoders, while others use learnable 1D absolute position embedding in ViT. Detailed information about RoPE and the models can be found in Appendix C, D.

## 3 INVESTIGATING OBJECT RECOGNITION IN VLMS

In this section, our goal is to understand how a model, from the shallower to deeper layers of the visual encoder [1] in a VLM, dynamically associates and combines tokens that belong to the same object but are positionally discontinuous in the image sequence, as illustrated in Figure 1. Let us consider an image as input and its output $H^{V,l} = (x_1^l, ..., x_{N_V}^l) \in \mathbb{R}^{N_V \times D_V} (l \in [1, L_V])$ at each layer of the visual encoder, and assume there are $M$ main objects $(o_1, ..., o_M)$ in the image.

### 3.1 METHOD: LOGIT LENS FOR VISUAL TOKENS

It is a good way to visualize the geometry of visual representations via representation similarity, while relying solely on the metric in the linear space is insufficient. This is because the semantics within an individual visual token becomes increasingly complex in deeper ViT layers, due to the interaction between visual tokens via self-attention (see Figure 8 and 9 in Appendix E.1). Thus, the semantics of visual tokens cannot simply be investigated using metrics such as cosine similarity. Logit lens is introduced to study the model behaviors from the view of activations (nostalgebraist, 2020). It applies the unembedding matrix $W_U \in \mathbb{R}^{D \times |\mathcal{V}|}$ to an activation in order to extract semantic information from it, where $\mathcal{V}$ is the vocabulary of the language model. In VLM, we can apply $W_U$ directly to the image representations and get the text tokens corresponding to the image patches:

$$W^V = (w_1^V, ..., w_{N_V}^V) = \underset{w \in \mathcal{V}}{\arg\max} \left( Softmax \left( W_U[(v_1^{l'}, ..., v_{N_V}^{l'})] \right) \right) \tag{4}$$

where $w_i^V$ is the decoded text token for image patch $p_i$, and $H^{l'} = (v_1^{l'}, ..., v_{N_V}^{l'})$ is visual part in the output of the $l'$-th layer in the LLM. Neo et al. (2024); Jiang et al. (2024) found that the text tokens contain rich semantic information related to their corresponding image areas. To interpret image processing in details, we inspect ViT layer by layer. For the $l$-th ViT layer, we (1) delete the ViT layers after the $l$-th ViT layer and (2) apply logit lens to the outputs of the $l'$-th LLM layer. In practice, the value of $l'$ is 25 in LLaVA-1.5-7B and 32 in Qwen2.5-VL-7B, as the emergence of the meaningful tokens is the most significant in these layers (see Appendix E.2). Thus, we actually create a family of functions $\mathcal{F} = \{f_l\}_{l=1}^{L^V}$ that map the image patches $P$ to their corresponding text tokens ($f_l : P \to W^{V,l}$) and reflect the dynamic process of object detection via natural language.

To visualize the dynamic process, we propose two concepts: *token map* and *segmentation map*. The idea originates from emoji logit lens (Sonia Joseph, 2024) that maps image patches to emojis for visualization. Here a token map is an image where each grid is filled with its corresponding text token. An illustration of the token map is shown in Figure 2. Examples of the token maps in LLaVA-1.5 and Qwen2.5-VL are shown in Figure 14-16 and 17-19, respectively. As for the segmentation map, we create a set of keywords including the names of the main

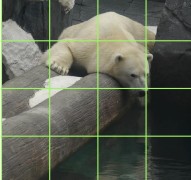

Figure 2: An illustration of token map.

elements in an object (e.g. for bear: {"bear", "head", "eye", "nose", "paw"}) for each of the main objects in the image. We fill in each grid with the color of object $o_m$ if the text token in that grid falls into the keywords set of object $o_m$. Details of this process are shown in Algorithm 1. The segmentation maps of LLaVA-1.5 are shown in Figure 3. See Appendix E.3 for more examples.

### 3.2 EXPERIMENTAL RESULTS AND ANALYSES

We use the samples from the GQA dataset (Hudson & Manning, 2019). As shown in Figure 3, the geometric structure in segmentation maps gradually approaches the shapes of the objects in the original image. We also observe that in the first few token maps of the earlier layers in ViT, tokens with no practical semantic meaning, such as punctuation or white spaces, account for the majority. From shallow to middle layers, attribute words for common local features (e.g. "fur", "yellow") begin to appear. While from middle to deep layers, the attribute words gradually disappear and the representative words (object labels) for specific global object (e.g. "bear", "rock") begin to

---

[1]The visual encoders in VLMs are typically based on the ViT architecture, so unless otherwise specified in the following text, we consider visual encoders and ViT as one thing.

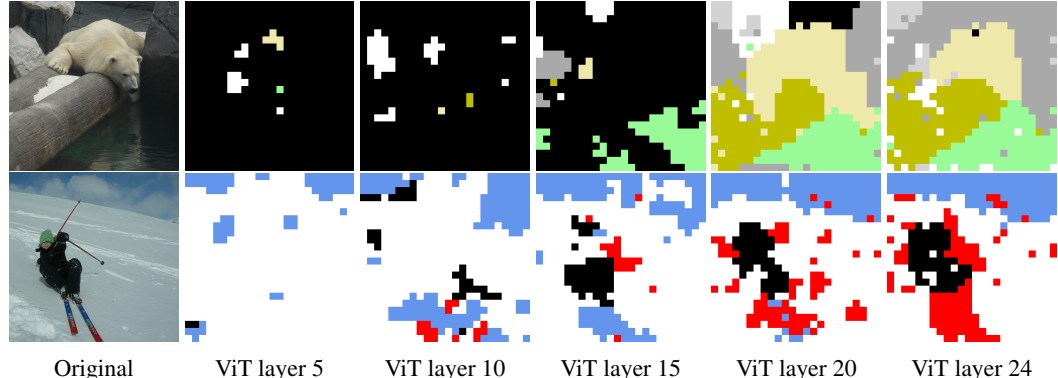

| Original | ViT layer 5 | ViT layer 10 | ViT layer 15 | ViT layer 20 | ViT layer 24 |

Figure 3: The segmentation maps of the image representations in ViT layers of LLaVA-1.5-7B. From shallow to deep layers, ViT gradually performs semantic disambiguation.

emerge. To quantify this phenomenon, we create a set of attribute words $\mathcal{W}^{A,o_m}$ and representative words $\mathcal{W}^{R,o_m}$ for each object $o_m$ in the image, and define the ratio of attribute words as $r_A = \frac{1}{N_V} \sum_{m=1}^{M} \sum_{w \in \mathcal{W}^{A,o_m}} \text{count}(w)$, where $\text{count}(\cdot)$ is the number of occurrences of a token. The ratio of representative words $r_R$ is defined in the same way.

We sample a batch of 50 images from GQA and compute $r_A$ and $r_R$ in each layer in LLaVA-1.5-7B. Results in Figure 4 indicate that the ratio of the attribute words starts to increase from layer 5, reaches its maximum at around layer 15, and then suddenly decreases. While at the same time, the representative words begin to appear in place of the attribute words. Therefore, the object recognition in ViT could be divided into two stages: *attribute recognition* (shallow to middle layers) and *semantic disambiguation* (middle to deep layers). In the first stage, attribute recognition is responsible for the detection of local low-level features such as color or texture, which could be shared among different objects. In the second stage, the model performs semantic disambiguation to integrate the common low-level features into specific high-level concepts.

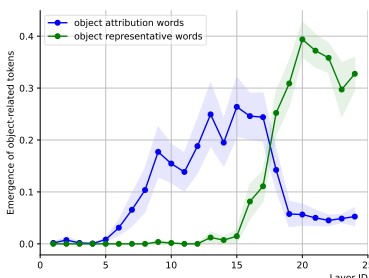

Figure 4: The change in the ratio of attribute words and representative words with ViT layers in LLaVA.

To further prove this, we evaluate the hallucination of LLaVA-1.5-7B on the POPE (Li et al., 2023) dataset, in which the questions are all in the form of "Is there a/an [object] ...?". We directly check layer by layer if any of the words in the token maps matches the [object] word. We set the answer to "yes" if any of the text tokens can be found in the synonyms set of the [object] from Lu et al. (2018), and "no" if not. Experimental settings are detailed in Appendix E.2. Results show that the accuracy hovers around 50% or less (random guess) until starting to increase at around layer 12, implying that the model has no confidence in what it sees until reaching middle layers. It first perceives discrete low-level features, and then integrate them together and view them as a whole, behaving in a way that follows the Gestalt principles of perceptual (Koffka, 2013) (Appendix E.4). By analogy, the model learns to link similar low-level features together via dot-product self-attention (Principle of Similarity and Proximity), and automatically fill in the "gaps" between discontinuous visual tokens via prior knowledge to view them as a whole object (Principle of Closure).

It should be noted that during pre-training, the parameters of the modality connector were adapted to the final layer of the ViT. Therefore, a more rigorous approach would be to train a dedicated modality connector for each ViT layer, though it is laborious to do this for each layer in each model. However, with the existence of the residual stream, every layer in a Transformer contains the output of all previous layers. This means that the last layer of a ViT is essentially a linear superposition of the current layer's output and the outputs of all previous layers. During VLM pre-training, on a macro level, the output from the last layer of the ViT is aligned with the language space via the modality connector. While from a micro perspective, the modality connector actually processes the last layer of the ViT, along with the output of all preceding layers. Therefore, it should have developed the ability to handle early ViT layers, and thus the reasonableness of our method is guaranteed.

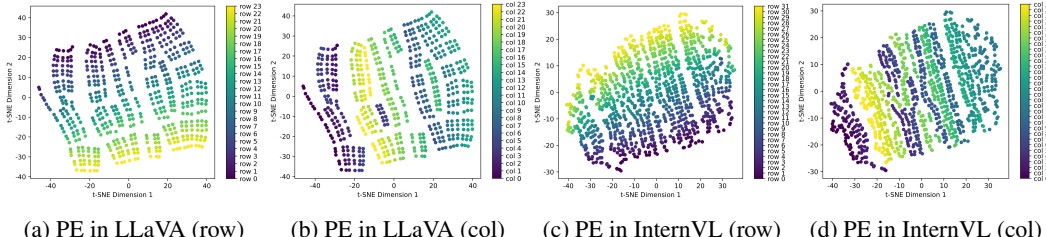

|  (a) PE in LLaVA (row) | (b) PE in LLaVA (col) | (c) PE in InternVL (row) | (d) PE in InternVL (col) |

Figure 5: The geometry of the 1D absolute position embedding in LLaVA-1.5-7B and InternVL-2.5-8B (w/o [CLS] token). The visualization is performed via t-SNE for dimensionality reduction. The labels are set to row IDs (5a, 5c) and column IDs (5b, 5d), respectively.

## 4 INVESTIGATING SPATIAL PERCEPTION IN VLMS

### 4.1 GEOMETRY STRUCTURE OF 1D ABSOLUTE POSITIONAL EMBEDDING

In this section, we explore how positional information is represented. Continuing the discussion in Section 2.2, we first analyze the learnable 1D absolute position embedding. This type of position embedding requires that the processed images have a fixed size. In LLaVA-1.5-7B, the position embedding $E \in \mathbb{R}^{N \times D^V}$ ($N = 577, D^V = 1024$) has a length of 577, corresponding to the size of a patchified 24x24 image (plus one [CLS] token). After pre-training, the position embedding for each patch should encode the unique row and column coordinate information for that position. To verify this, we apply t-SNE to reduce the dimensionality of the position embedding from LLaVA-1.5-7B and InternVL-2.5-8B to two dimensions. As visualized in Figure 5, the geometric structure of the absolute position embedding exhibits distinct rows and columns, which confirms our hypothesis.

However, we are more interested in 2D RoPE because it enables image processing with dynamic resolution and possesses better scalability. Unlike absolute position embedding, the image sequences processed by 2D RoPE are of variable length, making it impossible to assign a unique and fixed representation to each position. In this scenario, the position information can only be established upon the representations of objects and manifested through interactions between them. Therefore, it is an abstract and high-level feature, and we begin our analysis from a theoretical perspective.

### 4.2 THEORETICAL ANALYSES ON 2D ROPE

To simplify the discussion, we only consider two objects: A and B, and four spatial relationships between them: $\mathcal{R} = \{\text{"left"}, \text{"right"}, \text{"front"}, \text{"behind"}\}$. We simplify each object into a single patch, reduce the ViT dimensionality to 4 (minimum required for 2D RoPE), and ignore all other content in the image. Thus the shape of the image sequence is $2 \times 4$. As spatial relationships are relative, we use object B as a frame of reference and establish a coordinate system in the image where B is the origin, and the $X$ and $Y$ axes correspond to the directions of the image's width and height, respectively, as shown in Figure 6b. When considering the position of object A relative to object B, A and B can be termed the "satellite" and "nucleus," and the positions of object A corresponding to the four aforementioned relationships are $(-m, 0)$, $(m, 0)$, $(0, n)$ and $(0, -n)$, respectively.

We now investigate the representation of object A in these different positional relationships. Since RoPE is applied to queries and keys, we only consider the self-attention in ViT and ignore the scale factor, the Softmax function as well as the output projection for the convenience of discussion. When object A is to the left of object B, the attention output of object A in a layer can be written as:

$$h_A^{left} = <f(q_A, -m, 0), f(k_A, -m, 0) > v_A + <f(q_A, -m, 0), f(k_B, 0, 0) > v_B$$
$$= Re[q_A^X k_A^{X*} + q_A^Y k_A^{Y*}]v_A + Re[q_A^X k_B^{X*} e^{i(-m\theta)} + q_A^Y k_B^{Y*}]v_B \tag{5}$$

where $q_{(\cdot)}$, $k_{(\cdot)}$ and $v_{(\cdot)}$ denote the query, key and value, respectively, and all other notations are derived from Section 2.2. Similarly, we can obtain the representations for objects A and B across all four spatial relationships, $(h_A^r)_{r \in \mathcal{R}}$ and $(h_B^r)_{r \in \mathcal{R}}$ (Equation 24-31). It is obvious that they are weighted sums of $v_A$ and $v_B$, where the weighting coefficients are precisely where RoPE exerts its influence. Thus, we focus on the constituent components of these coefficients. First, by comparing the four representations of object A, $(h_A^r)_{r \in \mathcal{R}}$, we find that the coefficients of $v_A$ are iden-

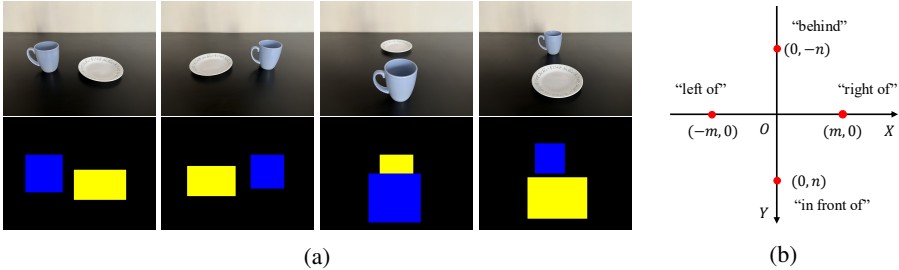

(a)                  (b)

Figure 6: (a): Examples in the What's Up B dataset. In the top row are original images, while the bottom row shows how we view each image. (b) The coordinate system in our theoretical analysis.

tical across all of them. The only difference lies in the $X$-axis related component of the $v_B$ coefficient. Comparing $h_A^{left}$ and $h_A^{right}$, we find that the $X$-axis components of the $v_B$ coefficient, $Re[q_A^X k_B^{X*} e^{i(-m\theta)}]$ and $Re[q_A^X k_B^{X*} e^{i(m\theta)}]$, formally possess a pair of *conjugate symmetric* terms. When written in the real-valued form (Equation 32, 33), they reveals a pair of *collinear and opposing* vectors: $\pm[(q_A^{(0)} k_B^{(1)} - q_A^{(1)} k_B^{(0)}) sin(m\theta)] v_B$. This determines why "left" and "right" are opposites in the model's visual geometry. Similarly, comparing $h_A^{left}$ and $h_A^{behind}$, we find that the $X$-axis and $Y$-axis components of their respective $v_B$ coefficients carry the relative positional information for the horizontal and vertical directions via different dimensions ($q_A^X k_B^{X*} e^{i(-m\theta)}$ vs. $q_A^Y k_B^{Y*} e^{i(-n\theta)}$). This implies the *orthogonality* of "left-right" and "front-back" in the model's view, and we will verify this through empirical studies.

Therefore, when the model is asked, "In which direction is A relative to B?", the components related to object B within object A's representation becomes the basis for determining the direction. Thus, the model should be able to determine the spatial relationship using only object A's representation, which will be verified later. However, recent studies on information flow in VLMs (Zhang et al., 2025) indicate that the LLM first attends to specific image regions based on the objects mentioned in the instruction, then aggregates the information into the last token for next-token prediction. Therefore, we further consider the interaction between objects A and B and propose the concept of the *direction vector*: $v^r = h_{o_S}^r - h_{o_N}^r (r \in \mathcal{R})$, where $o_S$ and $o_N$ are the satellite and nucleus, respectively. The subtraction is used for two reasons: (1) Though attention uses weighted sum, subtraction maintains linearity and does not affect our conclusions when comparing different spatial relationships later on. (2) It removes common terms from the two object representations, amplifying their differences and facilitating subsequent experimental analysis. We first compute the direction vectors for "left" and "right" (Eq. 34, 35). By comparison, $v^{left}$ and $v^{right}$ can be written as $c_1 v_A + c_2 v_B - (c_3 v_A + c_4 v_B)$ and $c_1 v_A + c_2 v_B + (c_3 v_A + c_4 v_B)$, where the terms for distinguishing positions are $c_3 = (q_B^{(0)} k_A^{(1)} - q_B^{(1)} k_A^{(0)}) sin(m\theta)$ and $c_4 = (q_A^{(0)} k_B^{(1)} - q_A^{(1)} k_B^{(0)}) sin(m\theta)$ (Eq. 36, 37). The opposing vectors in $v^{left}$ and $v^{right}$ indicate the collinearity of their encoded directions. While for $v^{behind}$ the $c_3$ and $c_4$ terms in it are both zero, it gets the terms $(q_B^{(2)} k_A^{(3)} - q_B^{(3)} k_A^{(2)}) sin(n\theta)$ and $(q_A^{(2)} k_B^{(3)} - q_A^{(3)} k_B^{(2)}) sin(n\theta)$, which are absent in $v^{left}$. When comparing $v^{left}$ and $v^{behind}$ (Eq. 38, 39), we find that although they are not strictly orthogonal, their directional information is encoded in different dimensions in the orthogonal subspaces of the $X$ and $Y$ axes. Thus, the model is supposed to separate them easily, which will be verified later. However, the key terms for spatial reasoning (e.g. $c_3$, $c_4$) are relatively small compared with the common terms, which could potentially limit the model's ability for spatial reasoning, and a possible solution is presented in Section 5.2.

To verify the above analyses, we use the subset B of the What's Up dataset (Kamath et al., 2023) (Figure 6a) and Qwen2.5-VL-7B. First, we conduct an experiment where we "*erase*" objects from an image, i.e., we replace the embedding of an object with another and check the change in the model's prediction of directions. Experimental results are shown in Table 4. We observe that when the embedding of B is replaced with that of A (or vice versa), the token probability of the right answer only drops a little bit ($0.909 \rightarrow 0.858$ on average). This indicates that an object's visual embedding contains sufficient information to determine its spatial relationships with other objects. Then, we compute the direction vectors. For each direction in $\mathcal{R}$, we randomly select the visual tokens from the embeddings of A and B and compute $h_{o_S}$ and $h_{o_N}$ via mean-pooling for 100 times. Details can be found in Appendix F.3. We visualized the 400 vectors via PCA. Figure 7a demonstrates the

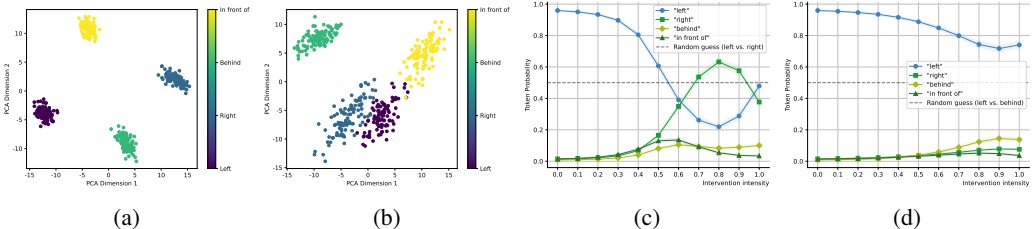

Figure 7: (a) and (b): The PCA results of the direction vectors for left of / right of / in front of / behind w and w/o position embedding. (c) and (d): Results of the intervention in visual embeddings demonstrate the collinearity of "left" & "right" and orthogonality of "left" vs. "in front of / behind".

collinearity and orthogonality of the direction vectors discussed before, while Figure 7b shows that the geometry of them is broken without 2D RoPE. Further, we intervene in the visual embeddings of A and B (where the relationship $r$ ="left") with those in $r'$ ="right" / "behind" via: $[V_o^r]' = (1 - \alpha)V_o^r + \alpha \operatorname{Mean}(V_o^{r'}) \cdot 1_{N_o}$, where $\alpha$ is the intervention intensity, $N_o$ is the number of patches in object $o$, $1_{N_o}$ is an all-ones vector in $\mathbb{R}^{N_o}$. After this intervention, the direction vector $v^r$ will become $(1 - \alpha)v^r + \alpha \cdot v^{r'}$ (see derivations in Equation 51). Results in Figure 7c and 7d show that the intervention from $V_o^{right}$ causes obvious performance drop as $\alpha$ increases, while $V_o^{behind}$ does not. This further verifies the geometry of positional information discussed before.

# 5 APPLICATIONS

## 5.1 TOKEN COMPRESSION BASED ON THE TOKEN MAP

Reducing the number of input visual tokens is an effective way for VLM inference acceleration. Current approaches are often based on similarity, which is insufficient (see Section 3.1), or are based on attention scores to measure instruction relevance, which is incompatible with efficient attention implementations. We focus on the inherent redundancy in an image and propose an instruction-agnostic method for token compression. Specifically, when the token map of an image is flattened into a 1D sequence of text tokens, we can observe many identical and positionally contiguous tokens within the sequence. Therefore, a natural idea is to compress the visual embeddings using the concept of run-length encoding (RLE) (Golomb, 1966). Our algorithm consists of two steps: (1) First, we count the consecutively repeated tokens and their repetition counts in the text token sequence of the visual embedding. (2) Second, for each run of consecutive tokens, we compress their corresponding embeddings into a single embedding, thereby shortening the image sequence length.

In practice, we introduce an additional "visual decoder" $\varphi$ between the modality connector and the LLM. It takes the visual embeddings as input and outputs its logits, which can be decoded into text tokens. It is initialized from the unembedding matrix $W_U$ in the LLM and trained via knowledge distillation (KD) (Hinton et al., 2015). Given a visual embedding $V = (v_1, ..., v_{N_V})$, it is processed by an $L$-layer LLM in the original model, resulting in the final logits $logits^{llm} = W_U[(v_1^L, ..., v_{N_V}^L)] \in \mathbb{R}^{N_V \times |\mathcal{V}|}$. The logits from the visual decoder is $logits^\varphi = \varphi[(v_1, ..., v_{N_V})] \in \mathbb{R}^{N_V \times |\mathcal{V}|}$. To make the distribution of $logits^\varphi$ closer to $logits^{llm}$, we adopt the loss for KD:

$$\mathcal{L} = \alpha\mathcal{L}_{soft} + (1 - \alpha)\mathcal{L}_{hard} = \alpha\tau^2 D_{KL}(P_T||Q_T) + (1 - \alpha)H(Y, Q) \tag{6}$$

where $\tau$ is the temperature, $Q_T = Softmax(\frac{logits^\varphi}{\tau})$ and $P_T = Softmax(\frac{logits^{llm}}{\tau})$ are the smoothed distributions for the student and the teacher, respectively. $Q = Softmax(logits^\varphi)$ is the actual output of the visual decoder $\varphi$, and $Y$ is the ground-truth labels derived from $Softmax(logits^{llm})$. The coefficient $\alpha$ controls the weight of the soft loss (KL divergence) and the hard loss (cross-entropy) for the learning of the distribution and the hard labels, respectively.

We test our method on LLaVA-1.5-7B and Qwen2.5-VL-7B. For the training of the visual decoder, we use the images from the train set of GQA and TextVQA (Singh et al., 2019), containing 148k and 22k samples, respectively. Training such a visual decoder is quite simple. As detailed in Appendix G, it takes approximately 5 hours on an A40 GPU and is highly data efficient. For LLaVA-1.5-7B, its visual decoder only requires training for $6k$ steps with a batch size of 16, using less than $100k$ of

Table 1: Evaluation results of the token compression method based on run-length encoding. The reduction rate is computed as the average reduction ratio on all samples during evaluation.

| Methods & Datasets | VQA$^{\text{v2.0}}$ | GQA | ScienceQA | TextVQA | MMBench$^{\text{EN}}$ | POPE | Reduction rate (%) |
|---|---|---|---|---|---|---|---|
| | | | Original Decoding | | | | |
| LLaVA-1.5-7B | 48.93 | 60.50 | 48.46 | 45.01 | 53.73 | 85.96 | / |
| Qwen2.5-VL-7B | 79.04 | 61.20 | 80.55 | 78.21 | 84.68 | 89.23 | / |
| | | | method1: Runlength compression | | | | |
| LLaVA-1.5-7B | 49.24 | 61.32 | 47.37 | 43.16 | 51.40 | 86.00 | 27.83 |
| Qwen2.5-VL-7B | 77.91 | 60.80 | 80.71 | 76.33 | 84.27 | 87.46 | 16.19 |
| | | method2: Runlength compression + Remove punctuation (Look at Top1 token) | | | | | |
| LLaVA-1.5-7B | 48.59 | 60.14 | 46.59 | 35.04 | 47.33 | 85.66 | 58.35 |
| Qwen2.5-VL-7B | 77.24 | 58.69 | 79.70 | 71.80 | 83.39 | 85.70 | 45.00 |
| | | method3: Runlength compression + Remove punctuation (Look at Top2 tokens) | | | | | |
| LLaVA-1.5-7B | 49.12 | 61.06 | 47.18 | 38.53 | 50.45 | 85.90 | 48.55 |
| Qwen2.5-VL-7B | 77.38 | 60.03 | 80.48 | 74.78 | 83.58 | 86.83 | 32.09 |

unlabeled data. For evaluation, we use GQA, TextVQA, VQA (Antol et al., 2015) and ScienceQA (Lu et al., 2022) for visual question answering. We also use MMBench (Liu et al., 2024b) and POPE for the test of the general capabilities and hallucination. As shown in Table 1, we implement three methods: (1) method 1 is the original algorithm; (2) method 2 deletes all the visual embeddings that are decoded into punctuation marks with a highest probability among all tokens in the vocabulary; (3) method 3 is based on method 2, while it only remove those with top-2 text tokens both being punctuation, which are more likely to be meaningless visual tokens. Details of these choices are shown in Algorithm 2. Further details of training and evaluation are presented in Appendix G.

Results show that our method could not only reduce sequence length for faster inference, but also control performance loss within an acceptable range. Compared with previous methods that are based on instruction relevance (Chen et al., 2024a; Zhang et al., 2024c; Lin et al., 2025; Zhu et al., 2024), though they seem to be better in terms of compression ratio and task performance, they are impractical in real-world scenarios because they cannot use FlashAttention (Dao et al., 2022) and require re-computing the relevance to the user prompt for every inference. Our method is more similar to the methods that focus solely on the image itself. The most representative is ToME (Bolya et al., 2022). While they measure the similarity between visual tokens via the keys from the ViT's attention, our method is more refined because it converts an image to natural language and merges similar tokens directly via text strings. Thus it has a lower compression ratio than ToME, but better performance on downstream tasks. More comparisons are provided in Appendix G.

## 5.2 ENHANCING SPATIAL REASONING WITH RECTIFIED RoPE

According to the discussions in Section 4.2, the component for positional discrimination (e.g., $\pm[(q_A^{(0)}k_B^{(1)} - q_A^{(1)}k_B^{(0)})sin(m\theta)]v_B$) has a relatively small magnitude compared with the common terms shared between the object representations of a pair of spatial relationships. This is because the values of trigonometric functions, which carry relative distance information, are always less than 1. To verify this, we focus on attention weights that RoPE directly influences. We split the dot-product attention into two parts: $q \cdot k^\top = (q^X, q^Y)(k^X, k^Y)^\top = q^X(k^X)^\top + q^Y(k^Y)^\top$, and compute the contribution to attention scores from the two axes by concatenating the two parts and applying Softmax to them together. For the $X$-axis and $Y$-axis, we separately select their "attention scores" from the "satellite" to the "nucleus" and average them across all attention heads and tokens (defined in Equation 53, 54). Details of this process are shown in Appendix H. The test is conducted on What's Up B and Qwen2-VL-2B. Results in Figure 23 show that for "left" and "right" (in the $X$-axis), attention from $Y$-axis is about 1.5 times larger than the X-axis, while the latter contains the key information for the determination of the positional relationship between two objects.

On the other hand, as the RoPE frequency $\theta_i = b^{-2i/d}$ decays rapidly when the dimension group index $i$ increases, the dimensions corresponding to large values of $i$ become highly insensitive to positional changes. For example, when $i = \frac{d}{2}$ and RoPE base $b = 10000$ (in Qwen2-VL), for relative distances of -50 and 50, the difference in their sine function values, $2sin(50\theta_{d/2})$, is merely around 0.01. These issues can all lead to critical information being "drowned out" by other irrelevant information when the model determines relative positional relationships between objects.

Table 2: Experimental results on spatial reasoning benchmarks for RoPE scaling and baselines.

| Methods & Datasets | What's Up A | What's Up B | VSR | COCO-spatial 1 | COCO-spatial 2 | GQA-spatial 1 | GQA-spatial 2 |
|---|---|---|---|---|---|---|---|
| Qwen2-VL-2B | 74.61 | 53.16 | 73.30 | 49.84 | 58.18 | 76.61 | 76.98 |
| Qwen2-VL-2B + RoPE scaling | 77.27 | 58.25 | 73.75 | 50.24 | 58.24 | 78.22 | 78.03 |
| Qwen2-VL-2B + SFT | 78.54 | 61.52 | 73.98 | 58.08 | **66.59** | 81.29 | 83.05 |
| Qwen2-VL-2B + SFT + RoPE scaling | **79.42** | **63.48** | **74.17** | 59.03 | **66.59** | **82.24** | **83.33** |
| Qwen2-VL-7B | 98.06 | 87.84 | 77.77 | 88.79 | 70.45 | 92.84 | 84.54 |
| Qwen2-VL-7B + RoPE scaling | 98.86 | 88.97 | 78.09 | 89.27 | 72.05 | 94.31 | 85.65 |
| Qwen2-VL-7B + SFT | 98.98 | 90.00 | 80.96 | 89.05 | 74.55 | 96.72 | **95.42** |
| Qwen2-VL-7B + SFT + RoPE scaling | **99.03** | **90.44** | **81.14** | **89.67** | **75.45** | **96.98** | 94.5 |

To address the above issues, we propose **RoPE scaling**, which adaptively scales the relative distances via a modification: $(\theta_i)' = \theta_i \cdot g(i)$, where

$$g(i) = 1 + \alpha(2i/d)^p \tag{7}$$

In Eq. 7, $d$ is the dimension of an attention head, $\alpha$ serves as the scaling magnitude, while $p$ is used to control the scaling factor $g(i)$ such that it applies minimal scaling for small $i$ and significant scaling for large $i$ (see Figure 24b for illustration). This compensates for the loss of positional information caused by frequency decay. We test our method (training-free / fine-tune on 60k samples from the GQA training set) on the Qwen2-VL series of models and evaluate on What's Up, VSR (Liu et al., 2023a), and COCO/GQA-spatial (Kamath et al., 2023). The selection of $\alpha$ and $p$, along with the training and evaluation setups, are presented in Appendix H. Results in Table 2 show that RoPE scaling achieves competitive performance across multiple benchmarks. We also test our method on MMBench and find that the model performance in general capabilities is even better with RoPE scaling, demonstrating its potential as a trick to enhance spatial reasoning ability. However, it serves only as a localized remedy for potential limitations in RoPE. We expect developments in positional encoding that can more effectively capture relative spatial relationships in the future.

## 6  RELATED WORK

**Interpretability of VLMs**   A line of work aims to interpret model components. For example, Goh et al. (2021); Gandelsman et al. (2023; 2024) find the neurons and attention heads in CLIP that correspond to real-world concepts. Elhage et al. (2022) demonstrate that neurons are polysemantic due to the existence of superposition, and sparse autoencoders (SAEs) are used to address this issue in LLMs (Bricken et al., 2023). Similarly, Hugofry (2024); Daujotas (2024); Rao et al. (2024) extract features in CLIP using SAEs. Similar to our work, Neo et al. (2024); Sonia Joseph (2024) interpret activations via logit lens. Sonia Joseph (2024) directly maps visual tokens to labels via the classification head in ViT. Neo et al. (2024) maps the visual tokens to texts via the language head. Our work is based on theirs, while we go a step further to systematically investigate VLMs layer by layer, and provide a detailed analysis of the dynamic visual information processing in a VLM.

Another line of work employs the idea of causal tracing originally developed for interpreting LLMs (Meng et al., 2022). Rajaram et al. (2024); Palit et al. (2023); Basu et al. (2024) investigate information storage and transfer in CLIP, BLIP and other VLMs via visual circuits. Recently, Neo et al. (2024); Zhang et al. (2024b; 2025) use blocking-based interventions to study the information flow in VLMs. Other approaches provide visualizations for the model internals. Some work provides heatmaps based on gradients (Selvaraju et al., 2017; Chefer et al., 2021; Zhang et al., 2024b), attention (Ben Melech Stan et al., 2024) or causal inference (Li et al., 2025).

## 7  CONCLUSION

We investigated image understanding in VLMs from the view of object recognition (the "what" way) and spatial perception (the "where" way). Based on the findings, we proposed an instruction-agnostic token compression method for inference acceleration, and RoPE scaling to improve the spatial reasoning capabilities of VLMs based on 2D RoPE. Our work offers new insights for both better understanding the mechanisms of visual information processing in VLMs and designing superior algorithms or model architectures. Limitations of this work are detailed in Appendix A.

## 8 ACKNOWLEDGEMENT

We thank the anonymous reviewers for their insightful comments. The research is supported by the Natural Science Foundation of Beijing, China (Grant No. L247010) and National Key R&D Program of China (Grant No. 2024YFF0907003).

## 9 REPRODUCIBILITY STATEMENT

To ensure reproducibility of this work, key supporting materials are distributed in the main text, appendix, and supplements. The anonymous source code for our proposed model/algorithm is available in Supplementary Material. For theoretical analysis for spatial perception, key assumptions and results are presented in Section 4 of the main text, with derivation process detailed in Appendix F.1. Dataset details—including sources, preprocessing, and splitting—are overviewed in the main text and detailed in the Appendix and Supplementary Material. These materials enable accurate replication of our results. Finally, LLMs are used to aid our writing of this paper.

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

## A  LIMITATIONS

To facilitate the future advancement of our work, we list some limitations here for discussion:

**(For object recognition in VLMs)**   In Section 3, while studying the object recognition process in VLMs, we directly extracted the output of a given ViT layer and passed it through the modality connector and language model, using a logit lens to examine the content represented by that layer. We discovered a two-stage process from attribute recognition to semantic disambiguation as the ViT layers deepen. However, it is possible that during pre-training, the parameters of the modality connector were specifically adapted to the final layer of the ViT. Directly passing outputs from other ViT layers through this connector may not fully reveal the content represented by those layers.

Therefore, we believe a more rigorous approach would be to add a dedicated modality connector for each ViT layer, fine-tune it on a large-scale dataset, and then observe the layer's representations. Though it is more rigorous, it would incur more computational costs and be more time-consuming and labor-intensive than our method. We thus leave this more rigorous approach for future work.

**(For spatial perception in VLMs)**   In Section 4, our study of spatial perception in VLMs only considered the four basic relationships of front, behind, left, and right. We did not (1) investigate the difference between "behind/above" and "in front of/below," nor did we (2) study more complex spatial relationships (e.g., top-left, bottom-right). Regarding (1), we believe a more complex structure exists within the model to distinguish between "behind" and "above" in a 2D image, which corresponds to 3D space. For instance, the model might observe the relative sizes of two objects and combine this with prior knowledge to infer depth properties and thus their spatial relationship. Regarding (2), we believe that based on our theoretical analysis in Section 4.2, more complex relationships like "top-left" could also be conveniently described by formulas and verified using the empirical methods from Section 4.2. We leave these investigations for future exploration.

## B  THE USE OF LARGE LANGUAGE MODELS

LLMs are used to aid our writing of this paper, primarily for checking vocabulary, correcting grammar, and polishing the prose.

## C  DETAILS FOR THE ROTARY POSITION EMBEDDING

### C.1  1D RoPE IN LANGUAGE MODELS

Rotary Position Embedding (RoPE) was originally introduced in language models to represent the relative distances between tokens (Su et al., 2024). Due to its excellent extrapolation capabilities, it is also adopted in vision-language models for position encoding in image sequences, enabling variable-resolution image processing. Here, we detail the working principle of RoPE and its practical implementation details.

First, we consider the 1D RoPE used in language models. The core idea of RoPE is to multiply each query and key by its positional information before calculating the attention scores. When a query and a key undergo a dot product, their positional information interacts and is converted into the relative distance between them. Specifically, RoPE can be expressed in both complex and vector forms. Suppose we have a two-dimensional query $q_m$ at position index $m$, with a vector form of $[q_m^{(0)}, q_m^{(1)}]$ and a complex form of $q_m^{(0)} + iq_m^{(1)}$. RoPE first applies a position-dependent transformation $f(q_m, m)$ to this vector to imbue it with positional information, as shown in Equation 8:

$$\begin{aligned} f(q_m, m) &= q_m e^{im\theta} \\ &= (q_m^{(0)} + iq_m^{(1)})(cos(m\theta) + isin(m\theta)) \\ &= [q_m^{(0)}cos(m\theta) - q_m^{(1)}sin(m\theta)] + i[q_m^{(0)}sin(m\theta) + q_m^{(1)}cos(m\theta)] \end{aligned} \tag{8}$$

Based on the result of Equation 8, we can express this transformation as a matrix multiplication, as shown in Equation 9:

$$
\begin{aligned}
f(q_m, m)^\top &= \begin{pmatrix} cos(m\theta) & -sin(m\theta) \\ sin(m\theta) & cos(m\theta) \end{pmatrix} \begin{pmatrix} q_m^{(0)} \\ q_m^{(1)} \end{pmatrix} \\
&= \begin{pmatrix} q_m^{(0)} cos(m\theta) - q_m^{(1)} sin(m\theta) \\ q_m^{(0)} sin(m\theta) + q_m^{(1)} cos(m\theta) \end{pmatrix}
\end{aligned}
\tag{9}
$$

As can be seen, the transformation RoPE applies to the query $q_m$ is effectively a multiplication by a rotation matrix that carries its positional information. The frequency is typically $\theta = b^{-2i/d}$, where $b$ is the RoPE base, $d$ is the model's dimension, and $i$ is the dimension group index (in this example, $i$ only takes the value of 1). Now, we consider the dot product between a query at position $m$ and a key at position $n$. First, we illustrate this process in Equation 10 using matrix multiplication:

$$
\begin{aligned}
&< f(q_m, m), f(k_n, n) > \\
=& f(q_m, m) f(k_n, n)^\top \\
=& \begin{pmatrix} q_m^{(0)} & q_m^{(1)} \end{pmatrix} \begin{pmatrix} cos(m\theta) & sin(m\theta) \\ -sin(m\theta) & cos(m\theta) \end{pmatrix} \begin{pmatrix} cos(n\theta) & -sin(n\theta) \\ sin(n\theta) & cos(n\theta) \end{pmatrix} \begin{pmatrix} k_n^{(0)} \\ k_n^{(1)} \end{pmatrix} \\
=& \begin{pmatrix} q_m^{(0)} & q_m^{(1)} \end{pmatrix} \begin{pmatrix} cos((m-n)\theta) & sin((m-n)\theta) \\ -sin((m-n)\theta) & cos((m-n)\theta) \end{pmatrix} \begin{pmatrix} k_n^{(0)} \\ k_n^{(1)} \end{pmatrix} \\
=& (q_m^{(0)} k_n^{(0)} + q_m^{(1)} k_n^{(1)}) cos((m-n)\theta) + (q_m^{(0)} k_n^{(1)} - q_m^{(1)} k_n^{(0)}) sin((m-n)\theta)
\end{aligned}
\tag{10}
$$

As can be seen, the dot product between the query and key effectively results in the fusion of their corresponding rotation matrices. The resulting rotation matrix contains their position difference, $m - n$, thereby modeling the relative distance. Similarly, based on the matrix multiplication form, we can express this in the form of complex multiplication, as shown in Equation 11:

$$
\begin{aligned}
&< f(q_m, m), f(k_n, n) > \\
=& f(q_m, m) f(k_n, n)^\top \\
=& Re[q_m e^{im\theta} \cdot (k_n e^{in\theta})^*] \\
=& Re[q_m k_n^* e^{i(m-n)\theta}] \\
=& (q_m^{(0)} k_n^{(0)} + q_m^{(1)} k_n^{(1)}) cos((m-n)\theta) + (q_m^{(0)} k_n^{(1)} - q_m^{(1)} k_n^{(0)}) sin((m-n)\theta)
\end{aligned}
\tag{11}
$$

As can be seen, when the query and key are expressed in complex form, their dot product is the real part of the multiplication between the transformed query and the conjugate of the transformed key. For the convenience of subsequent discussion, we will primarily use the complex form to represent the dot product process.

## C.2 2D RoPE in Vision-Language Models

Due to its favorable extrapolation properties, RoPE has played a positive role in long-text processing. To enable variable-resolution image processing in Vision-Language Models, Wang et al. (2024) introduced 2D RoPE. Similar to the basic form of 1D RoPE, 2D RoPE introduces two-dimensional positional information to separately represent information along the width and height directions for each patch in an image. Specifically, 2D RoPE divides the model's dimensions into two equal halves, used for processing information on the X-axis (width) and Y-axis (height), respectively. Suppose we have a four-dimensional query $q_m$ at coordinates $(m_1, m_2)$. Its vector form is $[q_m^X, q_m^Y] = [q_m^{(0)}, q_m^{(1)}, q_m^{(2)}, q_m^{(3)}]$, where $q_m^X = [q_m^{(0)}, q_m^{(1)}]$, $q_m^Y = [q_m^{(2)}, q_m^{(3)}]$. Its complex form is $[q_m^X, q_m^Y] = [q_m^{(0)} + iq_m^{(1)}, q_m^{(2)} + iq_m^{(3)}]$, where $q_m^X = q_m^{(0)} + iq_m^{(1)}$, $q_m^Y = q_m^{(2)} + iq_m^{(3)}$. The transformation $f(q_m, m1, m2)$ that 2D RoPE applies to this vector is shown in Equation 12:

$$
f(q_m, m_1, m_2)^\top = \begin{pmatrix} cos(m_1\theta) & -sin(m_1\theta) & 0 & 0 \\ sin(m_1\theta) & cos(m_1\theta) & 0 & 0 \\ 0 & 0 & cos(m_2\theta) & -sin(m_2\theta) \\ 0 & 0 & sin(m_2\theta) & cos(m_2\theta) \end{pmatrix} \begin{pmatrix} q^{(0)} \\ q^{(1)} \\ q^{(2)} \\ q^{(3)} \end{pmatrix}
\tag{12}
$$

When written in the complex form:

$$< f(q_m, m_1, m_2), f(k_n, n_1, n_2) > = Re\left[ \left( q_m^X e^{im_1\theta}, q_m^Y e^{im_2\theta} \right) \cdot \left( (k_n^X e^{in_1\theta})^*, (k_n^Y e^{in_2\theta})^* \right) \right]$$
$$= Re\left[ q_m^X k_n^{X*} e^{i(m_1-n_1)\theta} + q_m^Y k_n^{Y*} e^{i(m_2-n_2)\theta} \right]$$
(13)

It is clear that we have effectively constructed two separate rotation matrices, using $q_m^{(0)}$, $q_m^{(1)}$ and $q_m^{(2)}$, $q_m^{(3)}$ to represent positional information on the X and Y axes, respectively.

## C.3 ENGINEERING DETAILS OF RoPE

In practical implementations, there are two main styles for RoPE: GPT-J style and GPT-NeoX style. The GPT-J style pairs adjacent dimensions of the model to serve as the two dimensions for a rotation matrix, as shown in Equation 14:

$$f_{GPT-J}(x_m, m) = \begin{pmatrix} x^{(0)} \\ x^{(1)} \\ \vdots \\ x^{(d-2)} \\ x^{(d-1)} \end{pmatrix} \otimes \begin{pmatrix} cos(m\theta_0) \\ cos(m\theta_0) \\ \vdots \\ cos(m\theta_{d/2-1}) \\ cos(m\theta_{d/2-1}) \end{pmatrix} + \begin{pmatrix} -x^{(1)} \\ x^{(0)} \\ \vdots \\ -x^{(d-1)} \\ x^{(d-2)} \end{pmatrix} \otimes \begin{pmatrix} sin(m\theta_0) \\ sin(m\theta_0) \\ \vdots \\ sin(m\theta_{d/2-1}) \\ sin(m\theta_{d/2-1}) \end{pmatrix}$$
(14)

In contrast, the GPT-NeoX style, for computational efficiency, pairs $q_m^{(i)}$ $(i = 0, ..., (d-1)/2)$ with $q_m^{((d+1)/2+i)}$ through a "rotate half" operation, as shown in Equation 15:

$$f_{GPT-NeoX}(x_m, m)$$

$$= \begin{pmatrix} x^{(0)} \\ x^{(1)} \\ \vdots \\ x^{((d-3)/2)} \\ x^{((d-1)/2)} \\ \hline x^{((d+1)/2)} \\ x^{((d+3)/2)} \\ \vdots \\ x^{(d-2)} \\ x^{(d-1)} \end{pmatrix} \otimes \begin{pmatrix} cos(m\theta_0) \\ cos(m\theta_1) \\ \vdots \\ cos(m\theta_{d/2-2}) \\ cos(m\theta_{d/2-1}) \\ \hline cos(m\theta_0) \\ cos(m\theta_1) \\ \vdots \\ cos(m\theta_{d/2-2}) \\ cos(m\theta_{d/2-1}) \end{pmatrix} + \begin{pmatrix} -x^{((d+1)/2)} \\ -x^{((d+3)/2)} \\ \vdots \\ -x^{(d-2)} \\ -x^{(d-1)} \\ \hline x^{(0)} \\ x^{(1)} \\ \vdots \\ x^{((d-3)/2)} \\ x^{((d-1)/2)} \end{pmatrix} \otimes \begin{pmatrix} sin(m\theta_0) \\ sin(m\theta_1) \\ \vdots \\ sin(m\theta_{d/2-2}) \\ sin(m\theta_{d/2-1}) \\ \hline sin(m\theta_0) \\ sin(m\theta_1) \\ \vdots \\ sin(m\theta_{d/2-2}) \\ sin(m\theta_{d/2-1}) \end{pmatrix}$$
(15)

Most of the models nowadays (eg. Qwen2-VL) adopt the GPT-NeoX style for the implementation of RoPE, while we use the GPT-J style in our theoretical analysis for convenience of discussion.

# D  DETAILS FOR THE MODELS

In this work, we primarily use three models: LLaVA-1.5-7B, Qwen2 / 2.5-VL-7B, and InternVL-2.5-8B. We list the key parameters of these models relevant to our work in Table 3.

Table 3: Detailed information of the models used in this work.

|  | Model Info | LLaVA-1.5-7B | Qwen2.5-VL-7B | InternVL-2.5-8B |
|---|---|---|---|---|
| ViT | dynamic image size | False | True | True |
|  | image size | 336 | \ | 448 |
|  | patch size | 14 | 14 | 14 |
|  | num_hidden_layers | 24 | 32 | 24 |
|  | hidden size | 1024 | 1280 | 1024 |
|  | num_attention_head | 16 | 16 | 16 |
|  | head_dim | 64 | 80 | 64 |
|  | position embedding | 1D learnable (1+24*24) | 2D RoPE | 1D learnable (1+32*32) |
| Connector | downsampling rate | \ | 0.5 | 0.5 |
| LLM | num_hidden_layers | 32 | 28 | 32 |
|  | hidden_size | 4096 | 3584 | 4096 |
|  | vocab_size | 32064 | 152064 | 92553 |

# E  ANALYSIS BASED ON LOGIT LENS

## E.1  ANALYSIS ON THE REPRESENTATION SIMILARITY

For the analysis of the object recognition process in ViT, a natural idea is to use similarity-based methods for visualization. Therefore, for the output representation $H^{V,l} = (x_1^l, ..., x_{N_V}^l) \in \mathbb{R}^{N_V \times D_V} (l \in [1, L_V])$ of each ViT layer $l$, we first use t-SNE to reduce its dimensionality to three, and then perform clustering via K-means. The number of clusters, $K$, is set to the number of principal objects in the image (as provided in the GQA dataset). We employ two clustering strategies:

1. First perform standardization, then apply K-means clustering based on Euclidean distance (the conventional approach).
2. First perform L2 normalization on the representation of each image patch, then apply K-means clustering based on Euclidean distance.

The essence of the second clustering method is that it is actually based on the cosine similarity between data points. Specifically, after L2 normalization, the representation of each patch in the image sequence is a unit vector. At this point, the squared Euclidean distance between the representations of any two image patches can be written as: $||x_i - x_j||^2 = ||x_i||^2 - 2x_ix_j + ||x_j||^2 = 2(1 - x_ix_j)$. As can be seen, clustering based on Euclidean distance at this stage is effectively equivalent to clustering based on cosine similarity, since the term $x_ix_j$ is equal to the cosine similarity $\frac{x_ix_j}{||x_i||||x_j||}$ between unit vectors $x_i$ and $x_j$.

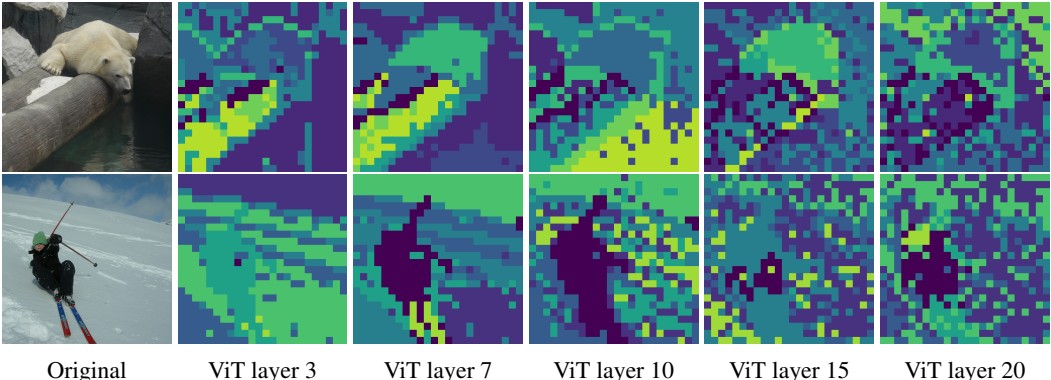

| Original | ViT layer 3 | ViT layer 7 | ViT layer 10 | ViT layer 15 | ViT layer 20 |

Figure 8: The visualization of the image representations in ViT layers of LLaVA-1.5-7B, based on dimensionality reduction (t-SNE) and clustering (K-means method 1).

The visualization results for the two clustering methods are shown in Figure 8 and 9, respectively. It is evident that as the ViT layers deepen, the clustering quality deteriorates. This is because in the shallower layers of the ViT, the image representations are still low-level visual features extracted during the initial patchifying process, resulting in clustering visualizations that resemble the original image. However, as the layers deepen, the features of the image patches undergo frequent interaction through the attention mechanism, and the representation at one position becomes mixed with representations from other positions. Furthermore, deeper representations often correspond to high-level semantic information, which is difficult to measure simply with linear metrics.

Recently, Siméoni et al. (2025) used a technique in DINOv3 called *Gram anchoring* to mitigate feature map degradation during self-supervised training by reducing the discrepancy between the student network's Gram matrix and that of an earlier model, thereby maintaining patch-level consistency. However, this method has not yet been validated in the training of VLMs. Therefore, studying model representations using metrics based on linear spaces, such as similarity, is often insufficient. We need to find more reliable metrics, such as the multimodal logit lens detailed in Section 3.

## E.2  DETAILS ABOUT THE TOKEN MAP

In Section 3.1, we provide an illustration of the token map and briefly describe the way of drawing a segmentation map based on the token map of an image. Here we discuss in detail about these.

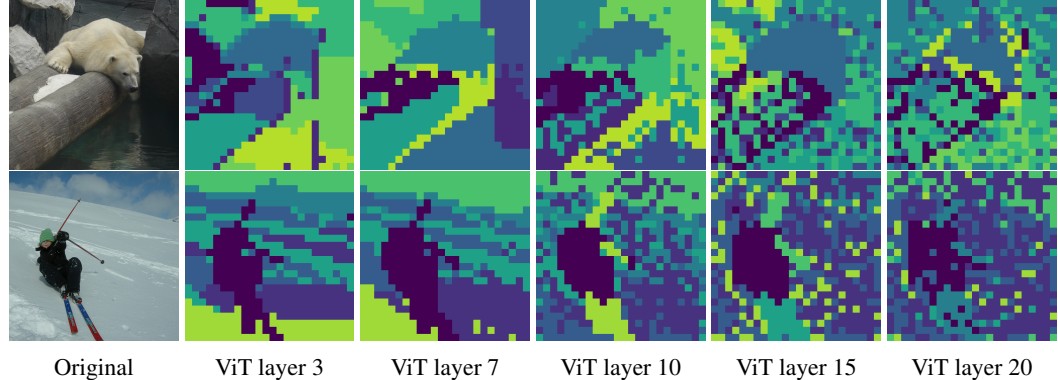

| Original | ViT layer 3 | ViT layer 7 | ViT layer 10 | ViT layer 15 | ViT layer 20 |

Figure 9: The visualization of the image representations in ViT layers of LLaVA-1.5-7B, based on dimensionality reduction (t-SNE) and clustering (K-means method 2).

The first step is to decide which layer's output from the VLM's language model should be used as the "logits" for the logit lens. Intuitively, the final layer might seem optimal, as it has been processed by more Transformer blocks and should thus contain richer information. However, this is not always the case. To make a more rigorous decision, we determine which layer's representation to use based on the emergence of *representative words*. Specifically, we first randomly select 500 samples from the GQA dataset. Each image in GQA is annotated with the names and bounding boxes of its principal objects, yielding an object name set $\mathcal{W}_*^o = \{w_*^{o_i}, ..., w_*^{o_m}\}$ for each image, where $w_*^{o_i}$ is the name or the "representative word" of object $o_i$. Then these samples enter the VLM, and we iterate through each layer of the language model. For each

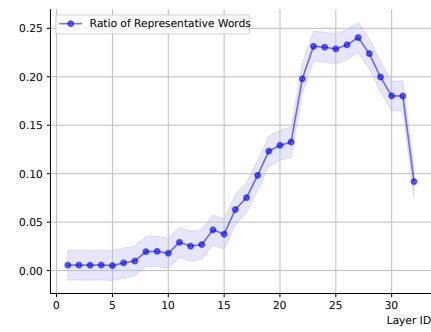

Figure 10: The emergence of representative words in LLaVA-1.5.

layer, we pass its output through the language model head to obtain the its logits, which are then decoded to generate a token map. For every text token in this map, if it falls within the object name set $\mathcal{W}_*^o$, we classify it as a representative word. After iterating through the entire token map, we calculate the emergence rate of representative words as $\frac{n_i^*}{N_V}$, where $n^*$ is the count of representative words and $N_V$ is the length of the image sequence.

We compute the average emergence rate for each layer of the language model, and the results for LLaVA-1.5-7B are presented in Figure 10. As can be seen, the 25th layer of LLaVA-1.5-7B shows significant emergence. Therefore, we select the representations from this layer for our subsequent experiments. Similarly, for Qwen2.5-VL-7B, we select the final layer.

In Figure 14, 15, 16 and 17, 18, 19, we present the token maps derived from LLaVA-1.5-7B and Qwen2.5-VL-7B, corresponding to the three images shown in Figure 11. The token maps are the results of the original model, without any intervention as described in Section 3.1. In token maps

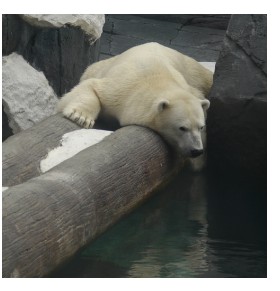
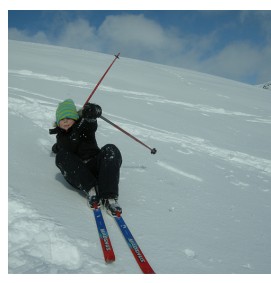
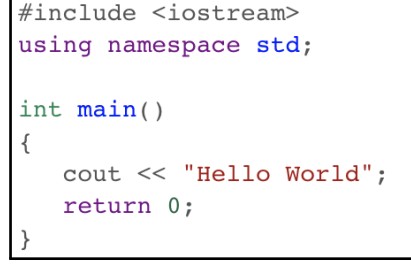

| (a) A bear on a log. | (b) A child skiing. | (c) C++ code. |

Figure 11: Example images in this section.

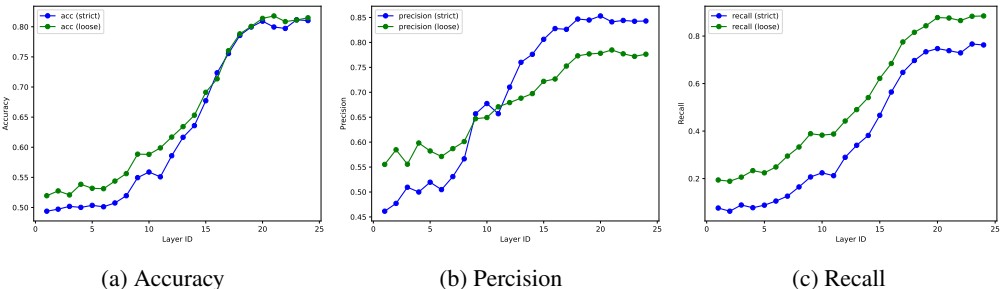

| (a) Accuracy | (b) Percision | (c) Recall |

Figure 12: Experimental results of LLaVA-1.5-7B on the POPE random set. The hallucination rate is tested directly using the token map corresponding to each ViT layer. From shallow to middle layers, the model is in a state of random guessing. While from middle to deep layers, the model gradually figure out the exact categories of the objects in the image (semantic disambiguation).

from LLaVA-1.5, we can observe a lot of meaningful text tokens that describe the image content in the corresponding image areas. Thus we provide a way of "*reading images like texts*". In token maps from Qwen2.5-VL, there are also lots of meaningful text tokens in both English and Chinese, since the model is pretrained on multilingual data with Chinese accounting for a large part of the corpus in the data. Besides, we also observe some meaningless tokens like the punctuation and whitespaces, or the "in" token which frequently appears in the token maps from LLaVA-1.5. We regard this kind of tokens as meaningless tokens that might be redundant for representing the image, which would be verified via our token compression method in Section 5.1.

In Section 3.2, we observe a tendency that the visual encoder exhibits a two-stage process from shallow to deep layers: it first performs attribute recognition from shallow to middle layers, and then performs semantic disambiguation from middle to deep layers to figure out the specific object categories in an image. To further demonstrate this, we test the model's hallucination layer by layer directly based on the token maps. Specifically, we use the POPE benchmark, in which the input to a model is an image together with a question asking whether a certain object is in that image or not. We use the POPE random test set and input images and questions into the VLM. Then we perform a layer-wise intervention on the ViT (visual encoder): we iteratively process up to each layer, discard all subsequent ViT layers, and then obtain the token map.

In the POPE dataset, the answer to each question is either "yes" or "no," indicating whether an object $o$ in the question is present in the image. We determine the final answer by checking if the token map contains the name of the target object. Specifically, the POPE dataset is built upon the MSCOCO dataset, and Lu et al. (2018) provide a synonym map for objects in MSCOCO. This allows us to treat semantically identical nouns as the same class (e.g., truck, pickup, lorry, hauler, firetruck all belong to the "truck" class). Therefore, we can iterate through all text tokens in the token map. If a token matches the target object's name or its synonym, the answer is set to "yes", otherwise it is "no".

Through this blocking intervention on the ViT layers, we obtain the accuracy, precision, and recall for each layer, as shown in Figure 12. Note that "strict" means that we only look at the Top-1 token (with the maximum probability), while "loose" means we look for the target object names in the Top-3 tokens. We observe that the VLM is essentially in a state of random guessing from the shallow to the middle layers of the ViT (up to layer 11). Starting around layer 11, the model's accuracy steadily improves. This indicates that the process of semantic disambiguation in the VLM truly begins in the middle layers, while the preceding layers are primarily processing low-level visual features, which corresponds to what we term the attribute recognition phase. Furthermore, the final hallucination rate measured via our token map approach (in terms of the three aforementioned metrics) is almost identical to the rate obtained from testing the model normally. This proves that the token map can faithfully represent the model's object recognition process.

### E.3 DETAILS ABOUT THE SEGMENTATION MAP

As for the segmentation map, it is bulit on the token map. The core idea is to identify the label (object category) of a text token in an image patch and fill the patch with the color of that label. The detailed process is shown in Algorithm 1. It generates a segmentation map through a two-phase process centered on context-aware token refinement. In the beginning, we write a set of keywords

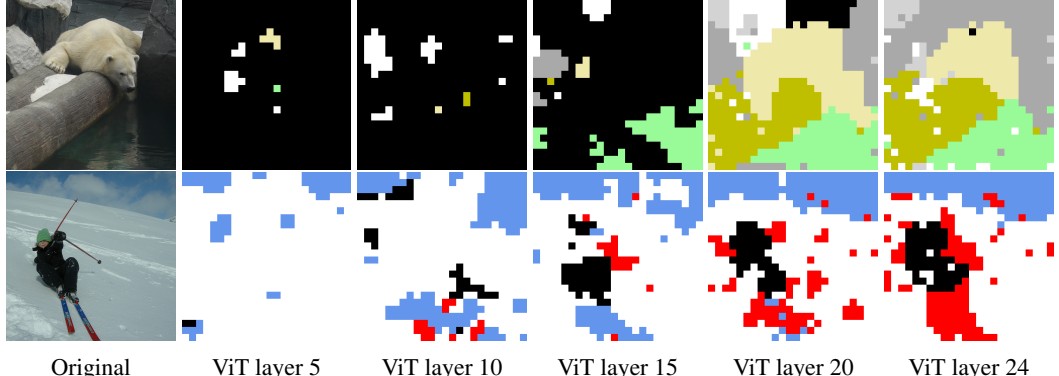

Figure 13: The segmentation maps of the image representations in ViT layers of LLaVA-1.5-7B. From shallow to deep layers, ViT gradually performs semantic disambiguation.

$\mathcal{W}^{o_i} = \{w_*^{o_i}, w_1^{o_i}, ..., w_n^{o_i}\}$ for each object $o_i$ of the main objects in an image. For example, for the object "bear" in Figure 11a, its keywords set is {"bear", "head", "eye", "nose", "paw"}, where "bear" is what we called the "*representative word* ($w_*^{o_i}$)" or the "*label*" of this object, while other keywords are either the aliases of the object, or the names of the elements in the object. Note that "background" and "others" are added as additional objects.

The first phase iteratively constructs a label map by evaluating each token from the initial token map. If a token is already a recognized keyword for a specific object, its corresponding object label is directly assigned to the new map. For unclassified tokens, the algorithm initiates a local refinement procedure. This procedure collects all meaningful neighboring tokens within a one-step Manhattan distance, filtering out any meaningless tokens such as punctuation or whitespaces. In the second phase, this completed label map is translated into the final visual output by assigning a predefined color to each position on the segmentation map according to its object label.

The core of the refinement lies in a voting mechanism used to determine the token's new identity. It first identifies the most frequent token(s) among the meaningful neighbors. If a unique winner is found by frequency, it is selected. In the event of a multi-way tie in frequency, the algorithm arbitrates by selecting the token whose occurrences have the minimum cumulative Manhattan distance to the central position, thereby prioritizing spatial proximity. The winning token's corresponding object label is then assigned to the map. If the winner is not a keyword for any known object, the label is set to "others". If a token has no meaningful neighbors, it is designated as "background".

### E.4 FURTHER DISCUSSIONS ABOUT THE GESTALT RECOGNITION

In Section 3.2, we discuss about the similarity between the object detection process in VLMs and human recognition. Here we put details about some of the Gestalt principles of perceptual, and discuss about the corresponding mechanisms in VLMs.

**Principle of Similarity:** We tend to group together elements that share similar visual characteristics, such as shape, color, or size. In VLMs, the scaled dot-product attention is related to representation (cosine) similarity. Thus the attention mechanism encourages the interaction between visual tokens, and allocates more attention to similar tokens.

**Principle of Proximity:** We tend to perceive elements that are close to each other as a single group. In VLMs, especially those using 2D RoPE, tokens that are positionally closer tend to have higher attention scores, and thus they are more likely to be in the same object in the view of VLMs.

**Principle of Closure:** Our minds automatically fill in gaps in an incomplete figure to perceive it as a whole, complete object. In VLMs, due to the raster scan operation, the image patches that are adjacent to each other in a 2D space may become positionally discontinuous in a 1D sequence. Nonetheless, the model is able to first associate them together as discussed in the aforementioned two principles, and then call prior knowledge stored in its weights to determine whether they belong to the same object. For example, though the feet of the child is not adjacent to his body in the image sequence in Figure 1, the model is still able to put them togther.

---

**Algorithm 1** Segmentation Map Generation via Token Refinement

---

**Require:** Object-keywords map $\{o \rightarrow \mathcal{W}^o\}$; Object-color map $\{o \rightarrow c\}$; Initial token map $Map^{token}$; Set of meaningless tokens $\mathcal{M}$.
**Ensure:** Final segmentation map $Map^{seg}$.

1: **function** RESOLVEUNCLASSIFIEDTOKEN($i, j, Map, \{o \rightarrow \mathcal{W}^o\}, \mathcal{M}$)
2:     $N \leftarrow [\,]$                     ▷ **Collect meaningful neighbors and their Manhattan distances**
3:     **for** $di \in \{-1, 0, 1\}$ **and** $dj \in \{-1, 0, 1\}$ **do**
4:         **if** $(di, dj) \neq (0, 0)$ **and** $(i + di, j + dj)$ is within map bounds **then**
5:             $neighbor \leftarrow Map[i + di, j + dj]$
6:             **if** $neighbor \notin \mathcal{M}$ **then**
7:                 Append $(neighbor, |di| + |dj|)$ to $N$
8:             **end if**
9:         **end if**
10:    **end for**
11:    **if** $N$ is empty **then return** "background"
12:    **end if**
13:    $F \leftarrow \text{CountFrequencies}(N)$; $C \leftarrow \text{FindMostCommon}(F)$
14:    $max\_count \leftarrow C[0].\text{count}$
15:    $TiedTokens \leftarrow \{t_k.\text{token for } t_k \text{ in } C \text{ if } t_k.\text{count} = max\_count\}$
16:    **if** $|\text{TiedTokens}| = 1$ **then**          ▷ **If there is a unique winner by frequency**
17:         $winner \leftarrow \text{TiedTokens}[0]$
18:    **else**          ▷ **Resolve multi-way ties by finding the one with minimum distance**
19:         $winner \leftarrow \text{FindTokenWithMinDistance}(TiedTokens, N)$
20:    **end if**
21:    $winner\_object \leftarrow \text{GetObjectNameForToken}(winner, \{o \rightarrow \mathcal{W}^o\})$
22:    **return** $winner\_object$ **if** $winner\_object \neq$ **null else** "others"
23: **end function**

24: **procedure** GENERATESEGMAP($Map^{token}, \{o \rightarrow \mathcal{W}^o\}, \{o \rightarrow c\}, \mathcal{M}$)
25:    $Map^{label} \leftarrow \text{NewMap}(h, w)$          ▷ **Phase 1: Refine to create a direct Label Map**
26:    **for** $i = 1$ **to** $h$ **do**
27:         **for** $j = 1$ **to** $w$ **do**
28:             $object \leftarrow \text{GetObjectNameForToken}(Map^{token}[i, j], \{o \rightarrow \mathcal{W}^o\})$
29:             **if** $object \neq$ **null then**
30:                 $Map^{label}[i, j] \leftarrow object$       ▷ **Directly assign the label if token is a keyword**
31:             **else**
32:                 $Map^{label}[i, j] \leftarrow \text{ResolveUnclassifiedToken}(i, j, Map^{token}, \{o \rightarrow \mathcal{W}^o\}, \mathcal{M})$
33:             **end if**
34:         **end for**
35:    **end for**
36:    $Map^{seg} \leftarrow \text{NewImage}(h, w)$             ▷ **Phase 2: Draw Segmentation Map**
37:    **for** $i = 1$ **to** $h$ **do**
38:         **for** $j = 1$ **to** $w$ **do**
39:             $Map^{seg}[i, j] \leftarrow \text{GetColor}(Map^{label}[i, j], \{o \rightarrow c\})$
40:         **end for**
41:    **end for**
42:    **return** $Map^{seg}$
43: **end procedure**

---

| | | | | | | | | | | | | | | | | | | | | | | | |
|---|---|---|---|---|---|---|---|---|---|---|---|---|---|---|---|---|---|---|---|---|---|---|---|
| in | in | and | and | rock | in | stone | pill | in | exhib | zoo | exhib | . | . | in | . | in | . | . | in | in | en | walls | exhib |
| and | wall | . | in | . | . | rock | wall | windows | zoo | and | . | . | . | . | . | . | . | . | . | in | stone | stone | rock |
| in | wall | rock | . | wall | wall | rock | . | polar | and | en | walk | in | in | in | and | . | in | . | walk | pill | in | . | gran |
| exhib | wall | wall | and | rock | rock | of | rock | exhib | side | . | on | bear | fur | bear | bear | in | Hinweis | ish | in | rock | wall | quar | polar |
| in | walls | wall | . | stone | rock | rock | in | on | bear | cub | bear | bear | bel | bel | bear | bear | on | in | of | . | stone | stone | stone |
| polar | columns | walls | in | in | . | . | on | in | in | bear | bear | . | bel | bear | bear | on | bear | lay | . | rock | . | rock | wall |
| in | . | blocks | in | wall | wall | on | on | bear | bear | bear | bear | bel | bear | bear | bear | cub | on | in | in | rocks | . | in | rock |
| in | blocks | stone | stone | p | on | in | bear | and | bear | bear | bear | bear | ears | bear | bear | bear | sleep | in | rocks | . | rocks | rock | rock |
| and | in | in | in | wall | extended | Cla | Cla | on | ing | bear | bear | bear | bear | ears | face | bear | bear | ears | . | rock | kwiet | rock | wall |
| in | stone | stone | wall | led | on | cla | cla | cla | on | on | on | bear | bear | bear | bear | bear | bear | cub | . | . | rock | . | rock |
| exhib | in | in | logs | logs | led | ra | in | . | wall | on | led | . | on | on | bear | expression | face | . | . | . | in | rock | in |
| . | logs | logs | logs | zoo | paint | walls | concrete | wall | led | logs | in | logs | logs | on | bear | and | face | on | . | . | . | . | rock |
| polar | logs | in | kwiet | paint | walls | walls | div | logs | in | logs | in | in | logs | logs | on | sleep | black | lips | . | . | . | rock | . |
| logs | zoo | logs | white | walls | white | logs | in | in | in | logs | logs | anch | in | in | logs | . | . | on | hair | in | . | . | infl |
| in | in | logs | white | logs | white | in | in | in | logs | in | in | kwiet | logs | logs | logs | on | on | in | . | . | . | . | in |
| concrete | logs | pipe | in | logs | and | in | polar | in | in | in | in | logs | logs | logs | logs | pool | water | in | . | . | . | . | water |
| in | logs | � | in | logs | zoo | logs | in | in | in | in | in | in | ro | logs | . | in | in | water | . | . | . | . | rock |
| logs | in | in | in | in | in | in | logs | in | logs | in | logs | ro | in | water | in | water | water | water | in | water | . | . | in |
| in | logs | in | logs | in | in | anch | polar | logs | in | logs | ra | water | in | polar | . | . | water | polar | water | " | logs | in | water |
| polar | log | in | in | | logs | in | . | in | logs | ✿ | pool | water | | water | mur | | in | reflection | in | in | in | mur | in |
| in | logs | in | in | in | logs | in | log | logs | pool | water | in | reflection | in | in | water | pool | in | in | in | in | black | in | water |
| in | polar | logs | in | in | logs | logs | pool | water | alg | | in | in | reflection | water | in | rock | reflection | in | in | mur | in | in | in |
| logs | a | logs | in | in | logs | water | in | mur | fish | fish | in | reflection | mur | | reflection | water | | in | in | | in | in | water |
| polar | in | polar | logs | wall | wall | polar | in | in | polar | reflection | water | in | polar | polar | in | pl | in | water | mur | in | a | in | polar |

Figure 14: The token map of Figure 11a from LLaVA-1.5-7B.

| sky | sky | to | in | clouds | . | . | . | sky | in | . | Hinweis | . | . | blue | . | . | . | clouds | | sky | clouds | sky | sky |
|---|---|---|---|---|---|---|---|---|---|---|---|---|---|---|---|---|---|---|---|---|---|---|---|
| sky | . | sky | sky | blue | cloud | cloud | iness | cloud | . | clouds | blue | cloud | lets | . | . | . | sky | . | . | . | . | . | . |
| in | in | cloud | in | . | cloud | cloud | cloud | . | cloud | . | . | . | . | sky | . | . | . | cloud | . | . | . | sky | sky |
| | ski | ski | . | in | Hinweis | snow | | | in | and | . | cloud | . | ski | . | cloud | . | cloud | in | . | . | . | cloud |
| snow | snow | . | . | . | in | . | ste | in | anten | and | . | | cloud | . | in | ski | cloud | cloud | cloud | in | . | cloud | sky |
| ste | snow | . | view | . | . | . | ste | ski | pole | . | . | . | ste | . | . | ste | sky | clouds | . | cloud | . | cloud | peak |
| ski | ste | . | in | . | . | snow | in | in | in | snow | ste | . | . | . | tracks | . | ste | ste | ski | | sky | in | in |
| snow | kwiet | ski | ste | in | snow | hat | handle | in | ski | . | . | ste | . | . | snow | sky | snow | . | snow | snow | in | ste | in |
| in | kwiet | snow | down | hat | hat | hat | gesture | . | ski | snow | . | snow | . | ( | . | ski | tracks | . | . | . | . | ste | snow |
| ski | tracks | snow | hat | hat | hat | hat | glo | glo | in | . | . | . | . | . | . | . | snow | . | snow | . | . | snow | . |
| snow | snow | ski | hat | in | eyes | Label | gesture | gesture | ski | ski | . | ste | . | . | . | . | snow | . | . | snow | anch | . | in |
| view | of | ski | jack | jack | in | snow | glo | ski | ski | ski | ski | in | . | snow | . | ste | . | snow | . | ski | . | . | ski |
| ackground | ski | down | on | glo | jack | jack | jack | on | je | in | in | ski | handle | ste | in | snow | . | . | . | . | . | snow | snow |
| . | snow | snow | on | je | je | in | suit | je | snow | trou | . | in | ski | . | . | . | . | background | tracks | snow | snow | . | |
| ski | . | � | on | on | snow | snow | jack | trou | je | in | in | in | snow | snow | tracks | in | . | . | kwiet | ✿ | . | . | snow |
| . | snow | of | in | shadow | kne | snow | je | in | je | on | . | . | snow | snow | ski | ste | snow | . | . | tracks | . | . | snow |
| ski | . | ski | . | | on | on | boot | leg | je | on | ski | anch | . | . | . | in | snow | . | . | . | | . | snow |
| snow | snow | . | | slope | slide | on | ski | sho | ski | ski | ✿ | in | snow | child | . | . | . | kwiet | . | . | . | . | . |
| in | tracks | tracks | . | snow | snow | down | ski | ski | ski | sho | down | . | anch | snow | . | | snow | . | . | . | . | snow | snow |
| snow | snow | . | . | snow | . | ski | ski | print | ski | number | ski | snow | . | . | . | . | . | kwiet | . | ste | . | . | in |
| . | snow | snow | snow | child | . | ski | ski | in | ski | Blue | ski | in | . | . | . | child | snow | . | . | . | . | . | kwiet |
| . | snow | snow | . | snow | snow | . | ski | stri | ( | ski | ( | ski | arms | snow | snow | snow | . | . | . | . | . | . | . |
| in | snow | . | snow | snow | snow | prints | in | ski | ski | ski | stri | ( | in | . | . | snow | . | snow | . | . | . | . | snow |
| snow | in | snow | snow | snow | snow | snow | snow | red | red | snow | ski | and | ski | | red | in | snow | snow | snow | snow | snow | snow | snow |

Figure 15: The token map of Figure 11b from LLaVA-1.5-7B.

| in | in | in | in | in | in | in | . | ( | in | . | in | . | in | . | . | in | . | . | . | in | in | in | code |
|---|---|---|---|---|---|---|---|---|---|---|---|---|---|---|---|---|---|---|---|---|---|---|---|
| in | in | in | in | in | in | . | . | Bedeut | Bedeut | . | ream | Bedeut | . | . | . |  |  | in | in | code | in | in | in |
| in | in | in | in | code | in | stream | in | for | . | . | in | text | text | . | text | code | text | code | code | . | . | . | . |
| in | in | in | in | in | in | in | in | code | kwiet | . | ( | pur | . | in | code | code | code | . | in | in | . | . | . |
| in | in | in | plate | plate | ake | ake | cap | Außer | Außer | Außer | Außer |  | ( | code | code |  | . | . |  | code | code | . | . |
| name | code | name | name | in | in | . | in | code | and | . | . | pur | code | code | name | code | code | . | code | code | code | . | code |
| name | in | in | in | in | in | . | IO | in | code | . | . | . | . | . | . | code | . | . | . | . | . | . | in |
| in | in | in | in | in | . | in | in | code | . | . | . | . | . | . | . | . | . | . | . | . | . | . | in |
| name | in | in | in | in | std | space | name | . | . | . | kwiet | . | . | . | in | . | . | . | . | . | . | in | in |
| name | code | in | name | ( | in | code | code | code | in | . | kwiet | . | . | . | NAME | . | . | . | . | . | . | . | blue |
| name | stream | name | name | name | code | code | code | code | in | . | . | . | . | . | . | NAME | � | kwiet | . | . | . | . | . |
| name | in | code | code | code | code | code | code | code | . | . | . | . | . | . | . | in | . | . | . | . | . | . | . |
| name | in | in | in | in | in | . | in | code | . | . | . | . | . | . | . | in | . | . | . | . | . | . | in |
| in | in | in | in | in | main | in | in | . | . | . | . | . | . | . | . | . | . | . | . | . | . | . | code |
| for | in | in | blue | in | in | Main | NAME | . | code | code | kwiet | . | in | . | in | code | code | code | in | . | ✿ | in | . |
| Bedeut | Bedeut | for | in | in | code | in | in | code | name | name | world | world | world | world | Bedeut | text |  | in | wide | wide | wide | wide |  |
| Bedeut | Hello | Hello | blue | code | hello | hi | Hello | code | hello | hello | hello | world | world | world | world | world | text | world | wide | wide | wide | wide |  |
| in | in | in | in | in | in | in | in | code | . | . | text | in | . | . | text | text | . | . | . | . | in | world | world |
| in |  | to |  |  | Return | code | return | return | code | code | code | . | name | " | . | in | . | . | . | . | . | . | in |
| Return | Return |  |  |  | code | code |  | code | code | in | in | in | anch | main | in | in | " | . | . | . | . | hello | . |
| return | in | pur | in | pur | in | code | . | code | code | in | . | . | . | . | . | . | . | . | . | . | . | . | . |
| pur | pur | of | pur | of | of | . | . | code | . | pur | . | . | " | anch | . | . | . | . | . | . | . | . | . |
| pur | in | in | in | in | in | of | in | in | . | . | . | . | . | . | . | pur | in | in | in | . | pur | in |
| in | code | in | in | in | pur | in | in | in | in | in | in | in | in | in | in | in | in | code | in | in | in | in | a |

Figure 16: The token map of Figure 11c from LLaVA-1.5-7B. The original image is resized to 336*336 px, which results in a 24*24 sized token map.

| 0 | <\|im_end\|> | | - | , | , | , | 地 | 图片 | 地面 | p | image | cub | | afs | | | i |
|---|---|---|---|---|---|---|---|---|---|---|---|---|---|---|---|---|---|
| 墙 | enclosure | 假 | 2 | exhibit | , | , | in | enclosure | 地面 | 续 | 地面 | enclosure | p | sl | | walls | |
| 膏 | 模拟 | 岩 | cub | exhibit | 块 | 裂缝 | | | 毛 | 色 | 熊 | cub | sl | exhibit | b | | 岩 |
| 块 | 模拟 | 岩 | bear | | 毛 | on | 色 | cub | bear | 米 | bear | bear | | paper | long | bear | |
| 0 | 1 | 冰 | man | cre | 伸 | cub | polar | in | 身 | 米 | large | bear | creamy | 3 | walls | 岩 | enclosure |
| 石 | 假 | 块 | ice | on | 伸 | cub | bear | cub | bear | white | ear | bear | | a | 岩 | formations | |
| large | led | , | 1 | p | 掌 | on | bear | 攀 | 毛 | and | ピン | bear | and | | cre | | 1 |
| 1 | , | 横 | 黑 | 边缘 | 掌 | 掌 | 块 | a | on | bears | bear | eyes | | water | in | pool | pro |
| long | cabin | enclosure | pe | on | frost | on | and | near | 圆 | a | its | nose | 鼻 | looking | polar | 岩 | 3 |
| exhibit | -like | ch | on | with | log | of | cub | 圆 | 圆 | a | on | n | n | between | | den | swimming |
| leaning | 桩 | 1 | paint | and | log | , | on | edge | trunk | a | the | | | of | <\|im_end\|> | of | head |
| 1 | segment | natural | 3 | zoo | natural | polar | <\|im_end\|> | 的 | zoo | brown | in | | water | 水 | in | 水 | 探 |
| per | 纹 | weather | natural | zoo | sea | bear | black | cabin | 横 | cross | cub | dipping | bear | small | into | water | cub |
| enclosure | trunk | , | bear | , | 木 | edge | large | 色 | . | in | | . | . | , | 水 | swimming | 3 |
| lean | grain | huge | 's | 0 | tree | enclosure | water | blue | its | 深 | . | . | light | | photo | . | contrast |
| in | large | fur | ` | } | 1 | pool | 深 | zoo | near | water | blue | blue | of | , | and | . | 1 |
| large | across | large | over | b | edge | . | blue | blue | r | pool | 4 | dark | light | body | edge | 4 | 探 |
| 0 | 0 | 木 | 4 | 0 | on | 1 | use | edge | } | dark | g | 1 | enclosure | | 探 | rock | polar |

Figure 17: The token map of Figure 11a from Qwen2.5-VL-7B. Some of the meaningful Chinese characters in this token map: "地" / "地面" = "ground", "墙" = "wall", "岩" = "rock", "石" = "stone", "块" = "block", "裂缝" = "crack", "毛" = "fur", "米" = "rice", "冰" = "ice", "纹" = "stripe", "熊" = "bear", "身" = "body", "掌" = "paw", "鼻" = "nose", "伸" = "stretch", "攀" = "climb", "水" = "water", "深" = "deep", "木" = "log", "圆" = "round".

| | | | | | | | | | | | | | | | | | |
|---|---|---|---|---|---|---|---|---|---|---|---|---|---|---|---|---|---|
| 0 | <\|im_end\|> | | - | - | 蓝 | 朵 | , | 条 | , | 蓝 | - | and | ing | | box | | i |
| � | 蓝 | 朵 | 脸 | 的 | 坐在 | 朵 | <\|im_end\|> | ing | 朵 | 朵 | white | 晴 | 晴 | 3 | dressed | 晴 | 5 |
| 雪 | 雪 | , | resort | , | 朵 | poles | | poles | y | and | down | on | 朵 | 白云 | snowy | sky | on |
| | 覆盖 | 覆盖 | ing | . | 白 | 杆 | 杆 | 边 | 山 | , | resort | poles | , | resort | . | landscape | one |
| 1 | 块 | 斜 | ski | 1 | 天气 | 杆 | Junior | , | and | 1 | tracks | a | plant | 蓝 | 白云 | 白云 | 2 |
| winter | 地 | 直 | anie | 在 | 杆 | and | <\|im_end\|> | 从 | . | 斜 | A | run | , | pristine | background | 峰 | |
| 坐在 | 踏 | 帽子 | 毛 | 毛 | | 掌 | ing | 滑 | snow | lines | on | tracks | 覆盖 | | skiing | , | left |
| 坐在 | 陷 | | 条 | cap | 保暖 | 握 | 杆 | , | ing | poles | <\|im_end\|> | ing | snow | behind | grip | tracks | 初 |
| red | 针织 | 1 | | 羽 | | 滑 | 杆 | 杆 | ski | 纵横 | Verb | ing | . | 1 | gripping | top | 2 |
| \|im_end\|> | , | on | 皮 | , | , | black | 黑色 | two | ski | | 滑 | tracks | patterns | 斜 | sports | ier | 白 |
| very | 滑 | on | pants | on | , | on | | , | | ski | long | learning | 斜 | tracks | 2 | right | 3 |
| - | sk | . | 1 | 坐在 | , | , | 1 | ing | 降 | 3 | child | poles | mult | from | | , | gripping |
| 3 | blue | 滑 | child | on | 1 | , | | 御 | holding | one | a | winter | 条 | right | still | little | 4 |
| dr | , | , | . | , | red | 蓝 | red | on | snow | 3 | snow | is | snow | sitting | on | . | |
| 踏 | 蓬 | | on | on | on | blue | blue | . | in | pristine | taking | . | pristine | controlled | 4 | ing | be |
| - | powder | , | and | child | i | blue | ski | S | sk | and | 小孩 | feet | snow | be | <\|im_end\|> | sk | sk |
| ing | 滑 | sitting | 块 | 4 | Atomic | . | | us | US | | blue | cl | tracks | | blue | snow | on |
| , | uneven | and | child | | deep | , | blue | <\|im_end\|> | | 两 | with | on | foot | one | | 4 | [ |

Figure 18: The token map of Figure 11b from Qwen2.5-VL-7B. Some of the meaningful Chinese characters in this token map: "蓝" = "blue", "(云)朵" / "白云" = "cloud", "晴" = "sunny", "雪" = "snow", "白" = "white", "覆盖" = "cover", "山" = "mountain", "峰" = "peak", "斜" = "oblique", "天气" = "weather", "坐" = "sit", "帽子" = "hat", "毛" = "fur", "保暖" = "keep warm", "掌" = "hand", "滑" = "ski", "杆" = "pole", "握" = "hold", "针织" = "knit", "黑色" = "black", "条" = "stripe", "降" = "down", "踏" = "pedal", "小孩" = "child".

| ` | !/ | < | --- | < | < | | iostream | ostream | ream | ream | > | > | | > | <\|im_end | #include | C | C | The | | the |
|---|---|---|---|---|---|---|---|---|---|---|---|---|---|---|---|---|---|---|---|---|---|
| include | include | < | < | < | <iostream | iostream | iostream | stream | ream | > | > | > | :: | | :: | <\|im_end | > | | :: | } | 这段 |
| using | namespace | namespace | main | namespace | paces | paces | std | std | std | <\|im_end | > | std | :: | | ; | ; | <\|im_end | > int | #include | > | cpp |
| | color | namespace | output | C | purple | | this | iostream | using | purple | C | library | ; | | <\|im_end | > int | #include | header | to | C | program |
| | main | main | 1 | () | () | blue | | ; | main | main | main | > | () | | () | include | std | C | output | i | |
| <\|im_end | > main | main | () | () | () | ) | <\|im_end | > { | { | ` | C | C | program | output | from | cout | <\|im_end | > std | background | provided |
| | | blue | () | () | () | <\|im_end | > { | 片段 | {} | | library | | :: | | :: | <\|im_end | >input | C | <\|im_end | <\|im_end | <\|im_end |
| <\|im_end | > | | << | the | << | | std | std | Hello | ello | , | , | () | | class | " | " | standard | << | | " |
| at | cout | out | << | << | << | << | " | " | Hello | o | , | , | C | | " | " | " | << | <\|im_end | <\|im_end | <\|im_end |
| | cout | out | << | << | | <\|im_end | <\|im_end | > ; | Hello | font | World | " | string | | , | "; | "; | ; | <\|im_end | <\|im_end | > , |
| 高 | return | return | | | a | ; | | ; | <\|im_end | <\|im_end | > 传统 | " | > | #include | C | include | iostream | output | 传统 | > |
| <\|im_end | <\|im_end | > `< | | C | color | ` | | ; | <\|im_end | <\|im_end | > std | C | snippet | ; | :: | This | the | namespace | std | | when |
| <\|im_end | <\|im_end | > 这段 | namespace | << | using | std | std | std | std | a | C | # | ; | C | file | main | standard | the | " | C |

Figure 19: The token map of Figure 11c from Qwen2.5-VL-7B. Some of the meaningful Chinese characters in this token map: "片段" = "segment", "这段" = "this segment".

# F DETAILS OF THE ANALYSIS ON SPATIAL PERCEPTION

## F.1 THEORETICAL ANALYSIS

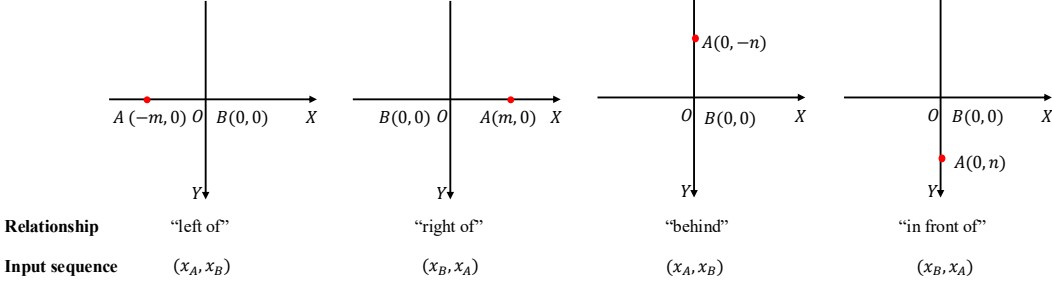

Figure 20: The idealized computational model for the theoretical analysis on spatial perception in VLMs. We choose four directions of A relative to B. The "Input sequence" refers to an input to a certain layer in the visual encoder. The order of A and B in the input sequence is the result of raster scan when the 2D image is flattened to a 1D sequence.

In Section 4.2, we theoretically analyzed how a VLM using 2D RoPE distinguishes between different opposing directions. Due to space constraints in the main text, we present some of the formulas and the detailed derivations here.

We consider two objects, A and B, simplifying each to a single image patch while ignoring the background and other content. The resulting input sequence has a length of 2, as illustrated in Figure 20. Considering a specific layer in the visual encoder, we first calculate the output of its attention module. In the attention calculation, we disregard the scaling factor, the Softmax operation, and the output projection, considering only the dot product between queries and keys and the weighted summation over values. We consider four directions $\mathcal{R} = \{$ "left", "right", "front", "behind" $\}$. When the direction of A relative to B is $r (r \in \mathcal{R})$, we denote the outputs for the positions of object A and

object B as $h_A^r$ and $h_B^r$, respectively. They can be expressed in a simple and clear form, as shown in Equations 16-23:

$$h_A^{left} = <f(q_A, -m, 0), f(k_A, -m, 0)> v_A + <f(q_A, -m, 0), f(k_B, 0, 0)> v_B \tag{16}$$

$$h_B^{left} = <f(q_B, 0, 0), f(k_A, -m, 0)> v_A + <f(q_B, 0, 0), f(k_B, 0, 0)> v_B \tag{17}$$

$$h_A^{right} = <f(q_A, m, 0), f(k_B, 0, 0)> v_B + <f(q_A, m, 0), f(k_A, m, 0)> v_A \tag{18}$$

$$h_B^{right} = <f(q_B, 0, 0), f(k_B, 0, 0)> v_B + <f(q_B, 0, 0), f(k_A, m, 0)> v_A \tag{19}$$

$$h_A^{behind} = <f(q_A, 0, -n), f(k_A, 0, -n)> v_A + <f(q_A, 0, -n), f(k_B, 0, 0)> v_B \tag{20}$$

$$h_B^{behind} = <f(q_B, 0, 0), f(k_A, 0, -n)> v_A + <f(q_B, 0, 0), f(k_B, 0, 0)> v_B \tag{21}$$

$$h_A^{front} = <f(q_A, 0, n), f(k_B, 0, 0)> v_B + <f(q_A, 0, n), f(k_A, 0, n)> v_A \tag{22}$$

$$h_B^{front} = <f(q_B, 0, 0), f(k_B, 0, 0)> v_B + <f(q_B, 0, 0), f(k_A, 0, n)> v_A \tag{23}$$

When we expand the dot product form into the complex form (see Equation 13), they can be written as shown in Equation 24-31.

$$h_A^{left} = Re[q_A^X k_A^{X*} + q_A^Y k_A^{Y*}] v_A + Re[q_A^X k_B^{X*} e^{i(-m\theta)} + q_A^Y k_B^{Y*}] v_B \tag{24}$$

$$h_B^{left} = Re[q_B^X k_A^{X*} e^{im\theta} + q_B^Y k_A^{Y*}] v_A + Re[q_B^X k_B^{X*} + q_B^Y k_B^{Y*}] v_B \tag{25}$$

$$h_A^{right} = Re[q_A^X k_A^{X*} + q_A^Y k_A^{Y*}] v_A + Re[q_A^X k_B^{X*} e^{im\theta} + q_A^Y k_B^{Y*}] v_B \tag{26}$$

$$h_B^{right} = Re[q_B^X k_A^{X*} e^{i(-m\theta)} + q_B^Y k_A^{Y*}] v_A + Re[q_B^X k_B^{X*} + q_B^Y k_B^{Y*}] v_B \tag{27}$$

$$h_A^{behind} = Re[q_A^X k_A^{X*} + q_A^Y k_A^{Y*}] v_A + Re[q_A^X k_B^{X*} + q_A^Y k_B^{Y*} e^{i(-n\theta)}] v_B \tag{28}$$

$$h_B^{behind} = Re[q_B^X k_A^{X*} + q_B^Y k_A^{Y*} e^{in\theta}] v_A + Re[q_B^X k_B^{X*} + q_B^Y k_B^{Y*}] v_B \tag{29}$$

$$h_A^{front} = Re[q_A^X k_A^{X*} + q_A^Y k_A^{Y*}] v_A + Re[q_A^X k_B^{X*} + q_A^Y k_B^{Y*} e^{in\theta}] v_B \tag{30}$$

$$h_B^{front} = Re[q_B^X k_A^{X*} + q_B^Y k_A^{Y*} e^{i(-n\theta)}] v_A + Re[q_B^X k_B^{X*} + q_B^Y k_B^{Y*}] v_B \tag{31}$$

As discussed in Section 4.2, by comparing the four representations of object A, $(h_A^r)_{r \in \mathcal{R}}$, we find that the coefficients of $v_A$ are identical across all of them. The only difference lies in the $X$-axis related component of the $v_B$ coefficient. Comparing $h_A^{left}$ and $h_A^{right}$, we find that the $X$-axis components of the $v_B$ coefficient, $Re[q_A^X k_B^{X*} e^{i(-m\theta)}]$ and $Re[q_A^X k_B^{X*} e^{i(m\theta)}]$, formally possess a pair of *conjugate symmetric* terms, as shown in Equation 32 and 33.

$$
\begin{aligned}
&Re[q_A^X k_B^{X*} e^{i(-m\theta)}] v_B \\
=&Re\big[(q_A^{(0)} + iq_A^{(1)})(k_B^{(0)} - ik_B^{(1)})\big(cos(m\theta) - isin(m\theta)\big)\big] v_B \\
=&Re\big[[(q_A^{(0)} k_B^{(0)} + q_A^{(1)} k_B^{(1)}) - i(q_A^{(0)} k_B^{(1)} - q_A^{(1)} k_B^{(0)})]\big(cos(m\theta) - isin(m\theta)\big)\big] v_B \\
=&\big[(q_A^{(0)} k_B^{(0)} + q_A^{(1)} k_B^{(1)})cos(m\theta) - (q_A^{(0)} k_B^{(1)} - q_A^{(1)} k_B^{(0)})sin(m\theta)\big] v_B
\end{aligned} \tag{32}
$$

$$
\begin{aligned}
&Re[q_A^X k_B^{X*} e^{i(m\theta)}] v_B \\
=&Re\big[(q_A^{(0)} + iq_A^{(1)})(k_B^{(0)} - ik_B^{(1)})\big(cos(m\theta) + isin(m\theta)\big)\big] v_B \\
=&Re\big[[(q_A^{(0)} k_B^{(0)} + q_A^{(1)} k_B^{(1)}) - i(q_A^{(0)} k_B^{(1)} - q_A^{(1)} k_B^{(0)})]\big(cos(m\theta) + isin(m\theta)\big)\big] v_B \\
=&\big[(q_A^{(0)} k_B^{(0)} + q_A^{(1)} k_B^{(1)})cos(m\theta) + (q_A^{(0)} k_B^{(1)} - q_A^{(1)} k_B^{(0)})sin(m\theta)\big] v_B
\end{aligned} \tag{33}
$$

As for the direction vectors, we first compare $v_{left}$ and $v_{right}$, as shown in 34 and 35.

$$
\begin{aligned}
v_{left} = h_A^{left} - h_B^{left} = &Re[q_A^X k_A^{X^*} + q_A^Y k_A^{Y^*} - q_B^X k_A^{X^*} e^{im\theta} - q_B^Y k_A^{Y^*}]v_A + \\
&Re[q_A^X k_B^{X^*} e^{i(-m\theta)} + q_A^Y k_B^{Y^*} - q_B^X k_B^{X^*} - q_B^Y k_B^{Y^*}]v_B
\end{aligned}
\tag{34}
$$

$$
\begin{aligned}
v_{right} = h_A^{right} - h_B^{right} = &Re[q_A^X k_A^{X^*} + q_A^Y k_A^{Y^*} - q_B^X k_A^{X^*} e^{i(-m\theta)} - q_B^Y k_A^{Y^*}]v_A + \\
&Re[q_A^X k_B^{X^*} e^{im\theta} + q_A^Y k_B^{Y^*} - q_B^X k_B^{X^*} - q_B^Y k_B^{Y^*}]v_B
\end{aligned}
\tag{35}
$$

Through comparison, it can be seen that $v_{left}$ and $v_{right}$ share a lot of common terms: $Re[q_A^X k_A^{X^*} + q_A^Y k_A^{Y^*} - q_B^Y k_A^{Y^*}]v_A$ and $Re[q_A^Y k_B^{Y^*} - q_B^X k_B^{X^*} - q_B^Y k_B^{Y^*}]v_B$. They can be written as $c_1 v_A + c_2 v_B$. The differences are underlined in Equation 36 and 37, which can be written as $-(c_3 v_A + c_4 v_B)$ and $(c_3 v_A + c_4 v_B)$, respectively. It is obviously that $v_{left}$ and $v_{right}$ share a pair of opposing vectors, implying the spatial geometry of the two directions.

$$
\begin{aligned}
v_{left} = &Re[q_A^X k_A^{X^*} + q_A^Y k_A^{Y^*} - q_B^Y k_A^{Y^*}]v_A + Re[-q_B^X k_A^{X^*} e^{im\theta}]v_A + \\
&Re[q_A^X k_B^{X^*} e^{i(-m\theta)}]v_B + Re[q_A^Y k_B^{Y^*} - q_B^X k_B^{X^*} - q_B^Y k_B^{Y^*}]v_B \\
= &Re[q_A^X k_A^{X^*} + q_A^Y k_A^{Y^*} - q_B^Y k_A^{Y^*}]v_A - (q_B^{(0)} k_A^{(0)} + q_B^{(1)} k_A^{(1)})cos(m\theta)v_A \\
&\underline{- (q_B^{(0)} k_A^{(1)} - q_B^{(1)} k_A^{(0)})sin(m\theta)v_A} + (q_A^{(0)} k_B^{(0)} + q_A^{(1)} k_B^{(1)})cos(m\theta)v_B \\
&\underline{- (q_A^{(0)} k_B^{(1)} - q_A^{(1)} k_B^{(0)})sin(m\theta)v_B} + Re[q_A^Y k_B^{Y^*} - q_B^X k_B^{X^*} - q_B^Y k_B^{Y^*}]v_B \\
= &c_1 v_A + c_2 v_B - (c_3 v_A + c_4 v_B)
\end{aligned}
\tag{36}
$$

$$
\begin{aligned}
v_{right} = &Re[q_A^X k_A^{X^*} + q_A^Y k_A^{Y^*} - q_B^Y k_A^{Y^*}]v_A + Re[-q_B^X k_A^{X^*} e^{i(-m\theta)}]v_A + \\
&Re[q_A^X k_B^{X^*} e^{im\theta}]v_B + Re[q_A^Y k_B^{Y^*} - q_B^X k_B^{X^*} - q_B^Y k_B^{Y^*}]v_B \\
= &Re[q_A^X k_A^{X^*} + q_A^Y k_A^{Y^*} - q_B^Y k_A^{Y^*}]v_A - (q_B^{(0)} k_A^{(0)} + q_B^{(1)} k_A^{(1)})cos(m\theta)v_A \\
&\underline{+ (q_B^{(0)} k_A^{(1)} - q_B^{(1)} k_A^{(0)})sin(m\theta)v_A} + (q_A^{(0)} k_B^{(0)} + q_A^{(1)} k_B^{(1)})cos(m\theta)v_B \\
&\underline{+ (q_A^{(0)} k_B^{(1)} - q_A^{(1)} k_B^{(0)})sin(m\theta)v_B} + Re[q_A^Y k_B^{Y^*} - q_B^X k_B^{X^*} - q_B^Y k_B^{Y^*}]v_B \\
= &c_1 v_A + c_2 v_B + (c_3 v_A + c_4 v_B)
\end{aligned}
\tag{37}
$$

As for the direction vectors of "left" and "behind", we analyze them in the same way as before, as shown in Equation 38 and 39. The underlined parts in the two equations represent the terms they do not contain relative to each other. To make it more clear, we re-write $v_{left}$ and $v_{behind}$ into Equation 40 and 41. We observe that the term $(k_3^Y v_A + k_4^Y v_B)$ in $v_{behind}$ is absent in $v_{left}$, while the term $(k_3^X v_A + k_4^X v_B)$ in $v_{left}$ is also absent in $v_{behind}$. We regard these terms as the key terms for determine the relationship between objects.

$$
\begin{aligned}
v_{left} = &Re[q_A^X k_A^{X^*} + q_A^Y k_A^{Y^*}]v_A + Re[-q_B^X k_A^{X^*} e^{im\theta} - q_B^Y k_A^{Y^*}]v_A + \\
&Re[-q_B^X k_B^{X^*} - q_B^Y k_B^{Y^*}]v_B + Re[q_A^X k_B^{X^*} e^{i(-m\theta)} + q_A^Y k_B^{Y^*}]v_B \\
= &Re[q_A^X k_A^{X^*} + q_A^Y k_A^{Y^*}]v_A + Re[-q_B^X k_B^{X^*} - q_B^Y k_B^{Y^*}]v_B + \\
&\big[ - (q_B^{(0)} k_A^{(0)} + q_B^{(1)} k_A^{(1)})cos(m\theta) - (q_B^{(2)} k_A^{(2)} + q_B^{(3)} k_A^{(3)}) \big]v_A + \\
&\underline{\big[ - (q_B^{(0)} k_A^{(1)} - q_B^{(1)} k_A^{(0)})sin(m\theta) \big]v_A +} \\
&\big[ (q_A^{(0)} k_B^{(0)} + q_A^{(1)} k_B^{(1)})cos(m\theta) + (q_A^{(2)} k_B^{(2)} + q_A^{(3)} k_B^{(3)}) \big]v_B + \\
&\underline{\big[ - (q_A^{(0)} k_B^{(1)} - q_A^{(1)} k_B^{(0)})sin(m\theta) \big]v_B}
\end{aligned}
\tag{38}
$$

$$
\begin{aligned}
v_{behind} =& Re[q_A^X k_A^{X^*} + q_A^Y k_A^{Y^*}]v_A + Re[-q_B^X k_A^{X^*} - q_B^Y k_A^{Y^*} e^{i(n\theta)}]v_A + \\
& Re[-q_B^X k_B^{X^*} - q_B^Y k_B^{Y^*}]v_B + Re[q_A^X k_B^{X^*} + q_A^Y k_B^{Y^*} e^{i(-n\theta)}]v_B \\
=& Re[q_A^X k_A^{X^*} + q_A^Y k_A^{Y^*}]v_A + Re[-q_B^X k_B^{X^*} - q_B^Y k_B^{Y^*}]v_B + \\
& \left[ -(q_B^{(0)} k_A^{(0)} + q_B^{(1)} k_A^{(1)}) - (q_B^{(2)} k_A^{(2)} + q_B^{(3)} k_A^{(3)})cos(n\theta) \right]v_A + \\
& \underline{\left[ (q_B^{(2)} k_A^{(3)} - q_B^{(3)} k_A^{(2)})sin(n\theta) \right]v_A} + \\
& \left[ (q_A^{(0)} k_B^{(0)} + q_A^{(1)} k_B^{(1)}) + (q_A^{(2)} k_B^{(2)} + q_A^{(3)} k_B^{(3)})cos(n\theta) \right]v_B + \\
& \underline{\left[ (q_A^{(2)} k_B^{(3)} - q_A^{(3)} k_B^{(2)})sin(n\theta) \right]v_B}
\end{aligned}
\tag{39}
$$

$$
\begin{aligned}
v_{left} =& v_{common} + (k_2^X v_B - k_1^X v_A)cos\phi + (k_2^Y v_B - k_1^Y v_A) \\
& - (k_3^X v_A + k_4^X v_B)sin\phi + 0 * (k_3^Y v_A + k_4^Y v_B)
\end{aligned}
\tag{40}
$$

$$
\begin{aligned}
v_{front} =& v_{common} + (k_2^X v_B - k_1^X v_A) + (k_2^Y v_B - k_1^Y v_A)cos\phi \\
& + 0 * (k_3^X v_A + k_4^X v_B) - (k_3^Y v_A + k_4^Y v_B)sin\phi
\end{aligned}
\tag{41}
$$

In Equation 40 and 41

$$
v_{common} = Re[q_A^X k_A^{X^*} + q_A^Y k_A^{Y^*}]v_A + Re[-q_B^X k_B^{X^*} - q_B^Y k_B^{Y^*}]v_B
\tag{42}
$$

$$
k_1^X = q_B^{(0)} k_A^{(0)} + q_B^{(1)} k_A^{(1)}
\tag{43}
$$

$$
k_2^X = q_A^{(0)} k_B^{(0)} + q_A^{(1)} k_B^{(1)}
\tag{44}
$$

$$
k_3^X = q_B^{(0)} k_A^{(1)} - q_B^{(1)} k_A^{(0)}
\tag{45}
$$

$$
k_4^X = q_A^{(0)} k_B^{(1)} - q_A^{(1)} k_B^{(0)}
\tag{46}
$$

$$
k_1^Y = q_B^{(2)} k_A^{(2)} + q_B^{(3)} k_A^{(3)}
\tag{47}
$$

$$
k_2^Y = q_A^{(2)} k_B^{(2)} + q_A^{(3)} k_B^{(3)}
\tag{48}
$$

$$
k_3^Y = q_B^{(2)} k_A^{(3)} - q_B^{(3)} k_A^{(2)}
\tag{49}
$$

$$
k_4^Y = q_A^{(2)} k_B^{(3)} - q_A^{(3)} k_B^{(2)}
\tag{50}
$$

## F.2 Empirical analysis on the representation of a single object

In Section 4.2, we first hypothesized that in an image sequence processed by the visual encoder, the representation of a single object contains sufficient information for the model to determine its spatial relationship with other objects. We then designed an experiment called "erasing objects" to test this hypothesis. The details are as follows.

"Erasing" an object in an image involves replacing the embeddings at positions corresponding to that object with the embeddings of another object within the image embedding $V \in \mathbb{R}^{N_V \times D}$, which has been processed by the visual encoder and modality connector. Specifically, following the simplifying assumption in Section 4.2, we consider an image containing only three elements: a satellite $o_S$, a nucleus $o_N$ and a background $o_B$, with corresponding position ID sets $\mathcal{I}_S$, $\mathcal{I}_N$ and $\mathcal{I}_B$ ($|\mathcal{I}_S| + |\mathcal{I}_N| + |\mathcal{I}_B| = N_V$) in the sequence of length $N_V$. For an image where the satellite's position relative to the nucleus is $r$, we define the embeddings for these three objects as $V_S^r = (v_j)_{j \in \mathcal{I}_S}$, $V_N^r = (v_j)_{j \in \mathcal{I}_N}$ and $V_B^r = (v_j)_{j \in \mathcal{I}_B}$. When we state that we "erase the nucleus by replacing it with the satellite," we replace all embeddings at positions $\mathcal{I}_N$ in the image embedding $V^r$ with the mean of the embeddings at positions $\mathcal{I}_S$, which is $\frac{1}{\mathcal{I}_S} \sum_{j=1}^{\mathcal{I}_S} V_{S,j}^r$. Other replacement cases are handled similarly and are not detailed here.

Table 4: Experimental results of "erasing objects". The notation "A → B" means erasing object A by replacing it with B. Results show that the representation within a single object is sufficient for the determination of the spatial relationship between the satellite and the nucleus.

| Methods & Models | Qwen2.5-VL-7B | | | | |
|---|---|---|---|---|---|
| | Left of | Right of | Behind | In front of | Overall |
| Original | **0.959** | **0.899** | **0.939** | **0.840** | **0.909** |
| Satellite → background | 0.905 | 0.848 | 0.934 | 0.676 | 0.841 |
| Nucleus → background | 0.949 | 0.897 | 0.889 | 0.782 | 0.878 |
| Satellite → nucleus | 0.907 | 0.850 | 0.924 | 0.640 | 0.830 |
| nucleus → Satellite | 0.943 | 0.886 | 0.837 | 0.764 | 0.858 |
| Satellite → background Nucleus → background | 0.897 | 0.850 | 0.884 | 0.583 | 0.804 |

In the experiment, the instruction we input to the VLM is: *"Is the {obj_satellite} in front of/behind/to the left of/to the right of the {obj_nucleus}? Please choose the best answer from the four options: [In front of, Behind, Left, Right], and reply with only one word. Your answer is: ".* To better observe changes in model performance, we calculate the sum of probabilities for the tokens corresponding to the correct answer in the output of the VLM. For example, if the correct answer is "left," we compute $p(\text{"left"}|x) + p(\text{"Left"}|x)$ as the model's confidence in the correct answer, where $x$ is the model's input. The experimental results are shown in Table 4. Evidently, the performance of the model does not suffer a significant loss when we apply various "erase objects" methods. This validates our initial theoretical analysis and hypothesis.

### F.3 EMPIRICAL ANALYSIS ON THE DIRECTION VECTORS

In Section 4.2, we formally defined the direction vector; here, we detail its calculation method. Contrary to our theoretical assumption, in a real image, an object does not correspond to just one image patch. As shown in Figure 6a, a single object can comprise tens to over a hundred image patches. In this case, we define the direction vector as $v^r = h_{o_S}^r - h_{o_N}^r = \text{Mean}(V_S^r) - \text{Mean}(V_N^r)(r \in \mathcal{R})$, where Mean represents mean pooling, and the definitions of $V_S^r$ and $V_N^r$ are detailed in Appendix F.2. In our actual calculations, to better validate our theory and visualize the direction vectors, we did not fix the representation $h_o^r$ of an object o to a single deterministic value $\text{Mean}(V_o^r)$. Instead, we randomly sample $n$ $(n < |V_o^r|)$ patch embeddings from $V_o^r$ and then perform mean pooling.

In the experiment, for a given spatial relationship $r$, we perform random sampling within the image patches of the satellite and nucleus to compute $h_{o_S}^r$ and $h_{o_N}^r$, respectively. We then subtract them to obtain the direction vector $v^r \in \mathbb{R}^D$. This process is repeated 100 times to yield 100 direction vectors for that relationship. The number of sampled image patches, $n$, is determined as follows: for objects with fewer than 20 patches, we do not perform sampling; for objects with more than 20 patches, we set $n$ to half the number of patches for that object.

Furthermore, in Section 4.2, we conducted an intervention experiment by using object representations from other spatial relationships to intervene on the object representations in the current relationship: $[V_o^r]' = (1 - \alpha)V_o^r + \alpha \text{Mean}(V_o^{r'}) \cdot 1_{N_o}$, where $r$ is the current relationship, $r'$ is another relationship, $\alpha$ is the intervention strength, and $1_{N_o}$ is a ones vector of length $N_o$ (i.e., $|V_o^r|$). In the experiment, we apply this intervention to both the satellite and the nucleus. The direction vector corresponding to the intervened image embedding is:

$$
\begin{aligned}
(v^r)' &= \text{Mean}([V_{o_S}^r]') - \text{Mean}([V_{o_N}^r]') \\
&= (1 - \alpha)\text{Mean}(V_{o_S}^r) + \alpha\text{Mean}(V_{o_S}^{r'}) - \left\{(1 - \alpha)\text{Mean}(V_{o_N}^r) + \alpha[\text{Mean}(V_{o_N}^{r'})]\right\} \\
&= (1 - \alpha)(\text{Mean}(V_{o_S}^r) - \text{Mean}(V_{o_N}^r)) + \alpha[\text{Mean}(V_{o_S}^{r'}) - \text{Mean}(V_{o_N}^{r'})] \\
&= (1 - \alpha)v^r + \alpha v^{r'}
\end{aligned}
\tag{51}
$$

## G   THE TOKEN COMPRESSION ALGORITHM BASED ON RUN-LENGTH ENCODING (RLE)

In Section 5.1, we propose a token compression method based on Run-Length Encoding (RLE). Prior methods primarily include those based on the similarity between visual tokens (Wen et al., 2025b; Alvar et al., 2025; Jeddi et al., 2025) and those based on instruction relevance (Chen et al., 2024a; Zhang et al., 2024c; Xing et al., 2024). Similarity-based approaches compress visual tokens according to their similarity. Such methods depend on the chosen similarity metric, and commonly used metrics like cosine similarity are not universally applicable. As discussed in Appendix E.1, in the deeper layers of a ViT, the representations of visual tokens often contain information from other tokens, and this entanglement becomes increasingly complex with depth. This makes it difficult to compute the similarity between visual tokens using simple metrics in a linear space.

As for methods based on instruction relevance, they measure the correlation between visual and text tokens based on attention scores and compress the visual tokens accordingly. However, this approach has significant practical drawbacks. First, modern models often rely on frameworks like FlashAttention (Dao et al., 2022) for fast inference, in which case the full attention score matrix is inaccessible, making such token compression methods difficult to implement. Second, Zhang et al. (2024a); Wen et al. (2025a) have pointed out that the existence of the "attention shift" phenomenon introduces bias when these methods measure relevance. More importantly, token compression methods guided by user instructions are often only applicable to the simplest question-answering datasets (e.g., VQA), and their utility in practical multi-turn dialogue scenarios is difficult to guarantee.

Therefore, building upon our interpretability analysis of VLM object recognition in Section 3.1, we propose an instruction-agnostic token compression method. The core of this method is the RLE concept, which reduces the length of the image sequence by compressing the visual embeddings (after the ViT and modality connector) before it enters the language model of the VLM. Specifically, we use the visual decoder proposed in Section 5.1 to decode the visual embeddings into text tokens, thereby obtaining a token map of the image. Then, we can read images like texts by treating consecutive and identical text tokens in the sequence as a group, which is a "run" in RLE. The corresponding image embeddings for this group of text tokens are then compressed into an embedding of length one via mean pooling or other methods. The specific algorithm is detailed in Algorithm 2.

For the training data, we randomly selected the first 148K samples from the GQA training set and added all 22K samples from the TextVQA training set to ensure the visual decoder has a certain degree of OCR capability. From this data, we randomly sampled 1,000 entries as a validation set. Here, we discuss the training details of the visual decoder. As shown in Equation 6, we train it via knowledge distillation, with the specific loss definition detailed in Section 5.1. During training, we

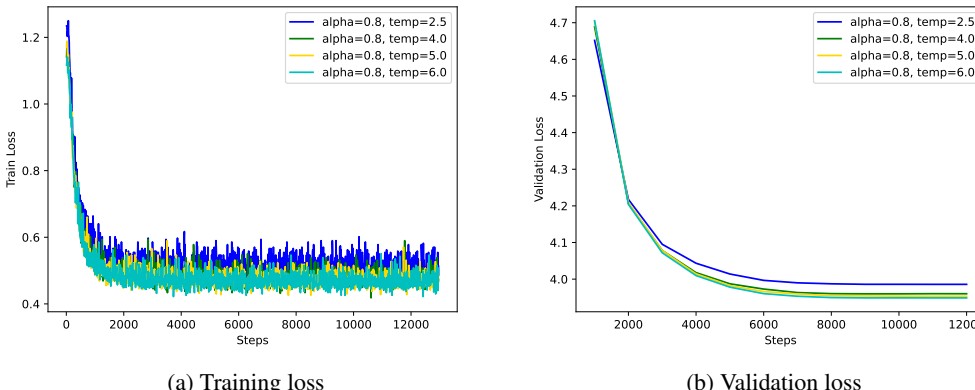

(a) Training loss                                    (b) Validation loss

Figure 21: The training and validation losses during the training of the visual decoder. Through comparison, we observe that a higher temperature leads to lower training and validation losses. When the temperature rises to around 5.0, the loss does not decrease significantly. Note that the validation loss only contains the hard loss to reflect the model's performance in practical applications.

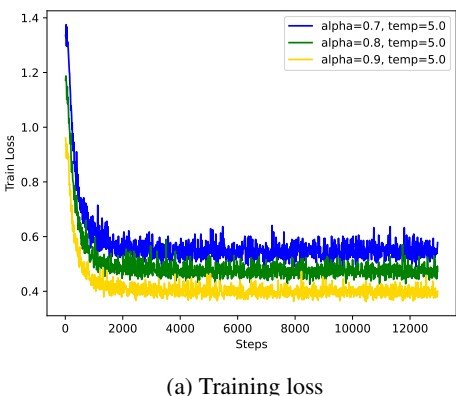 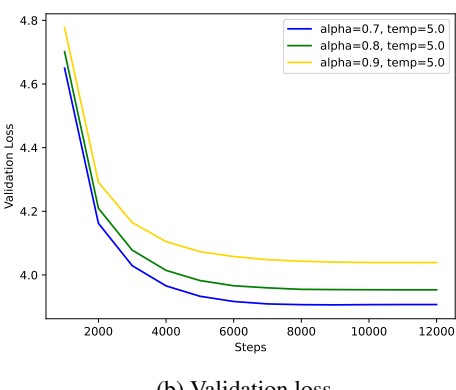

(a) Training loss                    (b) Validation loss

Figure 22: The training and validation losses during the training of the visual decoder. We observe that a higher loss scaling factor leads to lower training loss but higher validation loss. To strike a balance between ensuring the student model learns from the teacher's knowledge and maintaining its practical performance, we set $\alpha$ to 0.8 when training the visual decoder for Qwen2.5-VL-7B.

need to balance the soft loss (KL divergence between the target and original distributions) and the hard loss (cross-entropy loss between the target and original results) by adjusting the loss scaling factor $\alpha$. Concurrently, for the soft loss, we also need to adjust the distillation temperature $\tau$ to enable the student model to learn richer knowledge from the teacher model. We experimented with different hyperparameter combinations. For the temperature $\tau$, we tried values such as 2.5, 4.0, 5.0, and 6.0. A higher temperature results in a smoother probability distribution. We found that, with other conditions held constant, the loss gradually decreases as the temperature increases, but reaches a lower bound around a temperature of 5.0, as shown by the experimental results on Qwen2.5-VL-7B in Figure 21. For the final temperature setting, based on our experience, we use 2.5 for LLaVA-1.5-7B and 5.0 for Qwen2.5-VL-7B.

For the loss scaling factor $\alpha$, a larger value means a greater proportion of the soft loss. Compared to LLaVA-1.5-7B, Qwen2.5-VL-7B has a much larger vocabulary (152,064 versus 32,064), which means the supervision signal provided by the limited training data is sparser. Therefore, for Qwen, we appropriately increased $\alpha$ to help the model better learn the data distribution. As shown in Figure 22, we tried three values: 0.7, 0.8, and 0.9. During training, a larger $\alpha$ better helps the training loss to decrease. However, results on the validation set show that an overly large $\alpha$ leads to mediocre actual performance, as our validation set only considers the cross-entropy loss to reflect the model's performance in practical applications. Consequently, for the final setting of $\alpha$, we use 0.7 for LLaVA-1.5-7B and 0.8 for Qwen2.5-VL-7B.

Regarding other training parameters, for both models, we used a learning rate warmup strategy (warmup steps=1000) and a cosine annealing schedule. For LLaVA-1.5-7B, the learning rate was 1e-4 with a batch size of 16. For Qwen2.5-VL-7B, the learning rate was 2e-4 with a batch size of 32. Additionally, we employed an early stopping strategy, ending the training when the loss on the validation set did not decrease for 5 consecutive evaluations. LLaVA-1.5-7B was ultimately trained for 6,000 steps, and Qwen2.5-VL-7B for 9,000 steps.

In practical deployment, we adopted three approaches. The first is to compress all visual tokens. Since we observed many meaningless text tokens such as punctuation and spaces in the token map, we speculated that the semantics of these tokens are not very clear and may interfere with the model's prediction. Thus, in the second method, we remove all such tokens. The third method builds on the second, but requires that for a visual token to be considered meaningless, the top two most probable tokens decoded by the visual decoder must both be meaningless punctuation or spaces, rather than only considering the single most probable text token. As for the compression for a run of tokens in RLE, we randomly selected one visual embedding as the compression result, since we found that the performance of mean pooling is worse than random selection. The reason may be that the operation of mean pooling could hurt the original structure of visual embeddings.

For evaluation, we test on the benchmarks mentioned in Section 5.1. The evaluation for GQA is conducted by repeatedly sampling 1,000 samples from the test set 10 times and calculating the average of the accuracy scores. Furthermore, the truncation rate is defined as the average reduction ratio of the image sequence length during token compression, with the final ratio being the average result across all benchmarks.

For inference efficiency, we provide here a theoretical derivation for the compression ratio of our method. Since we only add an extra visual decoder after the modality connector compared to the original model, we ignore the computational cost of the ViT and the modality connector, and only consider the LLM. Furthermore, we focus only on the main overhead—the matrix multiplications within the Attention and MLP modules—while ignoring the text embedding, Softmax, activation functions, and the final language head.

Assume the LLM has $L$ layers, the model dimension is $d$, and the intermediate dimension of the MLP is $d'$. The visual decoder $\varphi$ has a size of $\mathbb{R}^{d \times |\mathcal{V}|}$, where $\mathcal{V}$ is the vocabulary. We consider the input of a single image, where the visual sequence length is $n$, and we ignore the length of the instruction. The computational cost of the original model is:

$$C = (8nd^2 + 4n^2d + 4ndd')L \quad \text{(FLOPs)}$$

When the compression ratio is $r$, the computational cost of our method is:

$$C' = [8n(1-r)d^2 + 4n^2(1-r)^2d + 4n(1-r)dd']L + 2nd|\mathcal{V}| \quad \text{(FLOPs)}$$

Therefore, we can measure inference efficiency by $\frac{C'}{C}$. For LLaVA-1.5-7B, the compression ratio $r$ of our method can reach $0.58$, corresponding to $\frac{C'}{C} = 43.82\%$, which is highly efficient.

---

**Algorithm 2** Token compression based on RLE (during the pre-filling stage)

---

1: **Input:** an image $X^V$, an instruction $X^T$, a VLM consisting of an visual encoder $f_I(\cdot)$, a modality connector $f_C(\cdot)$, and a language model $f_T(\cdot)$, a pretrained visual decoder $\varphi$.

2: **// — Get input embeddings and the text tokens for visual tokens —**
3: Get the visual embeddings $V = f_C(f_I(X^V) = (v_1, ..., v_{N_V}) \in \mathbb{R}^{N_V \times D}$ and text embeddings $T = (t_1, ..., t_{N_T}) \in \mathbb{R}^{N_T \times D}$;
4: Get the output of the visual decoder: $logits = \varphi[(v_1, ..., v_{N_V})] \in \mathbb{R}^{N_V \times |\mathcal{V}|}$;
5: Get the Top-1 text tokens: $W_1^V = (w_{1,1}^V, ..., w_{N_V,1}^V) = \underset{w \in \mathcal{V}}{\arg\max} \left( Softmax(logits) \right) \in \mathbb{R}^{N_V}$;
6: Get the Top-2 text tokens $W_2^V = (w_{1,2}^V, ..., w_{N_V,2}^V) \in \mathbb{R}^{N_V}$ according to the second highest probability, where $w_{i,2}^V = \underset{w \in \mathcal{V} \setminus \{w_{i,1}^V\}}{\arg\max} \left( Softmax(logits[i,:]) \right)$;

7: **// — Stage 1: Run-Length Encoding on all tokens —**
8: Define a set of meaningless tokens $\mathcal{M}$ (e.g., punctuation, whitespaces).
9: Initialize a list of runs $R \leftarrow [\ ]$, start index $s \leftarrow 1$, run length $l \leftarrow 1$.
10: **for** $i = 2$ **to** $N_V$ **do**
11:     **if** $w_i^V == w_{i-1}^V$ **then**
12:         $l \leftarrow l + 1$.                                    ▷ **Extend the current run**
13:     **else**
14:         Append $(s, l, w_{i-1,1}^V, w_{i-1,2}^V)$ to $R$.   ▷ **Save: (start, length, Top1 token, Top2 token)**
15:         $s \leftarrow i, l \leftarrow 1$.                                   ▷ **Start a new run**
16:     **end if**
17: **end for**
18: Append the last run $(s, l, w_{N_V,1}^V, w_{N_V,2}^V)$ to $R$.            ▷ **Save the final run**

19: **// — Stage 2: Filtering and Compression (Choose one method) —**
20: Initialize the compressed visual embeddings list $V' \leftarrow [\ ]$. Let $\text{Compress}(v_s, ..., v_{s+l-1})$ be a function that returns a single embedding (e.g., via random selection or mean pooling).

21: ---

22: **Method 1: Perform RLE-based compression on all visual tokens.**
23: **for** each run $(s, l, w_1, w_2)$ in $R$ **do**
24:     $\bar{v} \leftarrow \text{Compress}(v_s, ..., v_{s+l-1})$.
25:     Append $\bar{v}$ to $V'$.
26: **end for**

27: ---

28: **Method 2: Based on Method 1, but delete meaningless tokens (Top-1 based).**
29: **for** each run $(s, l, w_1, w_2)$ in $R$ **do**
30:     **if** $w_1 \notin \mathcal{M}$ **then**
31:         $\bar{v} \leftarrow \text{Compress}(v_s, ..., v_{s+l-1})$.
32:         Append $\bar{v}$ to $V'$.
33:     **end if**
34: **end for**

35: ---

36: **Method 3: Based on Method 2, but use Top-2 tokens for meaninglessness check.**
37: **for** each run $(s, l, w_1, w_2)$ in $R$ **do**
38:     **if not** $(w_1 \in \mathcal{M}$ **and** $w_2 \in \mathcal{M})$ **then**
39:         $\bar{v} \leftarrow \text{Compress}(v_s, ..., v_{s+l-1})$.
40:         Append $\bar{v}$ to $V'$.
41:     **end if**
42: **end for**

43: ---

44: **Output:** The input of the language model: $H^0 = (v_1', ..., v_M', t_1, ..., t_{N_T}) \in \mathbb{R}^{(M+N_T) \times D}$ ($M < N_V$), which is the combination of the compressed visual embeddings $V' = (v_1', ..., v_M')$ and the text embeddings $X^T$.

---

# H RoPE SCALING

In Section 4.2, we identify a potential deficiency in the spatial perception process of visual encoders based on 2D RoPE: the magnitude of the term responsible for distinguishing spatial relationships is often smaller than other terms. For a VLM, isolating this position-related representation from the multiple components of the image representation may be a non-trivial challenge. Furthermore, as the dimension group index $i$ of RoPE increases, its corresponding rotation angle $\theta_i$ decays exponentially, as shown in Figure 24a, leading to reduced sensitivity to positional changes.

To verify that the magnitude of the component responsible for representing spatial information is small relative to other components, we decompose the attention score calculation into two parts based on the $X$ and $Y$ axes. Assume the image sequence has length $n$ and the dimension of an attention head is $d$, where dimensions 1 to $\frac{d}{2}$ carry positional information for the $X$-axis, and dimensions $\frac{d}{2}+1$ to $d$ carry positional information for the $Y$-axis. The matrix multiplication of the query and key can be represented as in Equation 52:

$$
\begin{aligned}
QK^\top &= \begin{pmatrix} q_{11} & \cdots & q_{1d} \\ \vdots & \ddots & \vdots \\ q_{n1} & \cdots & q_{nd} \end{pmatrix} \begin{pmatrix} k_{11} & \cdots & k_{n1} \\ \vdots & \ddots & \vdots \\ k_{1d} & \cdots & k_{nd} \end{pmatrix} \\
&= \begin{pmatrix} q_{11} & \cdots & q_{1\frac{d}{2}} & q_{1(\frac{d}{2}+1)} & \cdots & q_{1d} \\ \vdots & \ddots & \vdots & \vdots & \ddots & \vdots \\ q_{n1} & \cdots & q_{n\frac{d}{2}} & q_{n(\frac{d}{2}+1)} & \cdots & q_{nd} \end{pmatrix} \begin{pmatrix} k_{11} & \cdots & k_{n1} \\ \vdots & \ddots & \vdots \\ k_{1\frac{d}{2}} & \cdots & k_{n\frac{d}{2}} \\ k_{1(\frac{d}{2}+1)} & \cdots & k_{n(\frac{d}{2}+1)} \\ \vdots & \ddots & \vdots \\ k_{1d} & \cdots & k_{nd} \end{pmatrix} \\
&= \begin{pmatrix} Q^X & Q^Y \end{pmatrix} \begin{pmatrix} K^X \\ K^Y \end{pmatrix} \\
&= Q^X K^X + Q^Y K^Y \\
&= M^X + M^Y
\end{aligned}
\tag{52}
$$

Here, $Q^X$ and $Q^Y$ correspond to the parts of the query that carry positional information for the $X$-axis and $Y$-axis, respectively. The same applies to the key. We then concatenate the dot product result from the $X$-axis dimensions, $M^X$, and the result from the $Y$-axis dimensions, $M^Y$, to obtain a dot product matrix $M^{(X,Y)}$ of shape $\mathbb{R}^{N \times 2N}$. We pass $M^{(X,Y)}$ through a Softmax operation to get the "attention score matrix" $A^{(X,Y)}$. By separating the parts corresponding to the $X$ and $Y$ axes, we obtain the "attention scores" contributed by the $X$-axis dimensions, $A^X$, and the $Y$-axis dimensions, $A^Y$. The "attention scores" we calculate reflect the relative contributions of the $X$-axis and $Y$-axis dimensions of the image representation to the final attention scores.

For two objects in an image, a satellite $o_S$ and a nucleus $o_N$, we focus on the "attention scores" from the satellite to the nucleus. Assuming the position IDs for these two objects in the sequence of length $N$ are $\mathcal{I}_S$ and $\mathcal{I}_N$, we calculate the "attention scores" for the $X$-axis and $Y$-axis parts from the satellite to the nucleus as follows:

$$
a_{S \to N}^X = \frac{1}{n_1 n_2 n_h} \sum_{h=1}^{n_h} \sum_{i \in \mathcal{I}_S, j \in \mathcal{I}_N} A_{ij}^{h,X}, \quad n_1 = |\mathcal{I}_S|, n_2 = |\mathcal{I}_N| \tag{53}
$$

$$
a_{S \to N}^Y = \frac{1}{n_1 n_2 n_h} \sum_{h=1}^{n_h} \sum_{i \in \mathcal{I}_S, j \in \mathcal{I}_N} A_{ij}^{h,Y}, \quad n_1 = |\mathcal{I}_S|, n_2 = |\mathcal{I}_N| \tag{54}
$$

In Equation 53 and 54, $n_h$ is the number of attention heads, and $A_{ij}^{h,X}$ and $A_{ij}^{h,Y}$ are the "attention scores" from query $q_i$ to key $k_j$ at the $h$-th attention, corresponding to the dimensions from the $X$-axis and $Y$-axis, respectively. In practice, we conduct experiments using the What's Up subset B. We perform this calculation for each layer $l$ of the ViT on all samples to obtain the layer-wise "attention scores" $a_{S \to N}^{X,l} = \frac{1}{K} \sum_{k=1}^{K} a_{S \to N,k}^{X,l}$ and $a_{S \to N}^{Y,l} = \frac{1}{K} \sum_{k=1}^{K} a_{S \to N,k}^{Y,l}$ (for $l \in [1, L_V]$), where $K$ is the number of samples in the dataset. Finally, for each ViT layer $l$, we normlize $a_{S \to N}^{X,l}$ and $a_{S \to N}^{Y,l}$ to $[0, 1]$. Results on Qwen2-VL-2B are shown in Figure 23. In Figure 23a and 23b, the

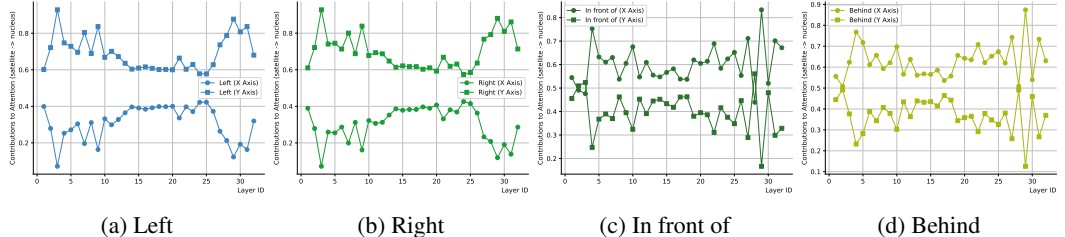

| (a) Left | (b) Right | (c) In front of | (d) Behind |

Figure 23: The contribution to attention scores from the dimensions in the $X$-axis and $Y$-axis.

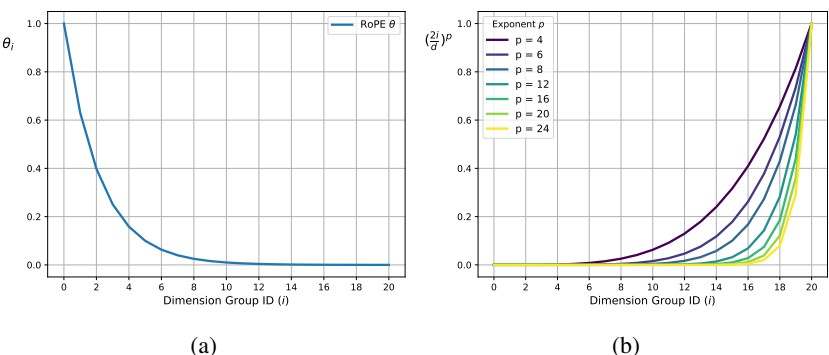

| (a) | (b) |

Figure 24: (a)The decay of RoPE $\theta$. (b)The compensation for the decay of RoPE $\theta$ via RoPE scaling.

difference in spatial relationships between the two objects lies in the $X$-axis, while the contribution to the final attention scores from the $X$-axis is smaller than that of the $Y$-axis. Similar phenomenon can be found in Figure 23c and 23d.

Therefore, we propose RoPE scaling. The core idea is to compensate for the decay of $\theta_i$ by amplifying the distance information in the low-frequency dimensions (corresponding to larger dimension group indices $i$). In other words, we want all dimensions to effectively represent positional information. In our experiments, we first conducted training-free experiments, in which we directly changed the original 2D RoPE to the RoPE scaling form in Equation 7. The training-free results shown in Table 2 are the best results obtained by tuning the hyperparameters $\alpha$ and $p$ for each benchmark.

To more accurately reflect the effect of RoPE scaling, we also fine-tuned both the original model and the model with RoPE scaling applied. The fine-tuning data consists of 60K samples related to spatial reasoning, randomly sampled from the GQA training set. We require that either the question or the answer should contain at least one word in the keyword list for spatial reasoning: ["left", "right", "top", "bottom", "in front of", "behind", "above", "below", "next to", "beside", "between", "on top of", "under", "over"]. During fine-tuning, all training settings, including batch size, learning rate, and training data, were kept identical to ensure a fair comparison, with the only exception being the RoPE scaling parameters. Specifically, for Qwen2-VL-2B, we performed full fine-tuning with a batch size of 32 and a learning rate of 3e-6. For Qwen2-VL-7B, we used LoRA fine-tuning with a batch size of 32, a learning rate of 3e-6, and a LoRA rank and alpha of 16 and 32, respectively. All models were trained for one epoch. For the models using RoPE scaling, we searched for the hyperparameters $\alpha$ and $p$, and the best results are shown in Table 2. The corresponding $\alpha$ and $p$ are 99 and 8 for Qwen2-VL-2B, and 49 and 8 for Qwen2-VL-7B.

We evaluated the fine-tuned models on the evaluation sets described in Section 5.2, and found that RoPE scaling demonstrates excellent spatial reasoning performance across multiple test sets, as shown in Table 2. To ensure the practical utility of RoPE scaling, we also evaluated the general capabilities on MMBench (English). The original performance (accuracy) of Qwen2-VL-7B on MM-Bench is 84.82%. We found that for the fine-tuned Qwen2-VL-7B, the accuracy of the model with RoPE scaling was 85.56%, which is even higher than the model without RoPE scaling (85.45%). Therefore, RoPE scaling, as a method for enhancing the spatial awareness of VLMs, can maintain the general capabilities of the model without degradation.

