# OpenReview forum: "Reading Images Like Texts: Sequential Image Understanding in Vision-Language Models"
_ICLR.cc/2026/Conference — ICLR 2026 Poster_

### Official Review · Reviewer_fdDb · 2025-10-25

**Soundness:** 3
**Presentation:** 3
**Contribution:** 4
**Rating:** 6
**Confidence:** 4

**Summary:**

The paper examines how vision-language models (VLMs) encode and reason with visual information within a one-dimensional token sequence. It addresses two core questions: how VLMs associate positionally discontinuous tokens representing the same object to identify its category ("what"), and how they infer 2D spatial relationships between objects ("where"). The authors use logit lens and visualizations to uncover a two-stage visual processing pattern. They theoretically and empirically analyze how RoPE-based encoders represent spatial relationships, revealing their underlying geometric structure. Building on these insights, the paper proposes two practical methods: a token compression approach that shortens image sequences with minimal performance loss, and a RoPE scaling technique that enhances spatial reasoning while maintaining overall model performance.

**Strengths:**

* The analysis section is thorough and compelling, offering valuable insights into how visual encoders and visual tokens are processed within VLMs. This contributes to a deeper understanding of the inner mechanisms driving object and spatial reasoning.

* The proposed applications are well grounded in the paper’s analytical findings and serve as effective validations of the authors’ observations, demonstrating both conceptual soundness and practical relevance.

* The theoretical and empirical analysis of RoPE-based spatial processing highlights a real limitation in the current VLM architecture. By identifying and addressing this limitation, the work provides a meaningful direction for future research and improvements in spatial reasoning within VLMs.

**Weaknesses:**

__Paper structure:__

The paper’s structure could be improved. Much of the core analysis is deferred to the appendix, making it difficult to follow the main findings from the main text alone. At the same time, several sections—particularly the Related Work—contain substantial repetition of content already covered in the Introduction. Streamlining the exposition and integrating key analyses into the main body would make the paper more cohesive and readable.

__Comparison to prior work__:

The paper lacks a sufficient comparison to other recent token compression methods, such as [1]. Since prior methods already identify salient tokens and achieve significant token reduction, it is unclear whether (1) the proposed method identifies similar or distinct subsets of tokens, and (2) whether it provides any advantage in compression efficiency or downstream task performance. A quantitative or qualitative comparison with these approaches would strengthen the paper’s claims.

__Training-free RoPE scaling experiment:__

In the training-free RoPE scaling experiment, the authors report the best results achieved by tuning the hyperparameters \alpha and p. It is unclear whether these parameters were optimized using the test set. If so, this constitutes data leakage and invalidates the experiment. The authors should clarify the procedure used to select these parameters (e.g., validation split, held-out set) and ensure that the evaluation remains unbiased.


[1] Kaduri et al., What's in the Image? A Deep-Dive into the Vision of Vision Language Models, CVPR 2025

**Questions:**

My only requests/questions on top of the previously mentioned points are about newer architectures and higher-dimensional RoPEs:

* Will this analysis transfer to models that use DeepStack [2], like Qwen3?

* Can you say something similar about 4D RoPEs in videos (i.e., improving the "where" pathway for video frames, or extending it to a "when" pathway)?

[2] Meng et al., DeepStack: Deeply Stacking Visual Tokens is Surprisingly Simple and Effective for LMMs, NeurIPS 2024

---

> ### Author Response · Authors · 2025-11-19
>
> Thanks a lot for your time on our work. I'll try my best to answer your concerns!
>
> **Weaknesses**
>
> **(1) Paper structure**
>
> We apologize for any inconvenience caused to your reading. We'll try to reorganize our paper!
>
> **(2) Comparison with other methods**
>
> Due to page limitations, we did not discuss about this.  For now, we would like to briefly compare our method with prior work.
>
> **Does our method identify similar or distinct subsets of tokens?**
>
> Prior methods are mainly based on instruction relevance, or the similarity between visual tokens. The first type of methods, such as the work you mentioned, as well as similar methods like Fast-V, SparseVLM, VisPruner, AdaFV, etc., use the attention and/or cosine similarity between the image and the text to measure the importance of visual tokens. Our method is different from these, because it focuses on the intrinsic redundancy of an image via logit lens. It ignores the user prompt. Therefore, compared to these methods, our method prunes different tokens from them.
>
> The second type is similar to our work, which focuses solely on the image itself. The most representative is ToME [1]. They measure the similarity between visual tokens via the keys from the ViT's attention. The visualization in that paper is very similar to our segmentation maps. Although our idea is different from it, we both utilize the semantics of the image content alone for token merging, and thus identify similar subsets of tokens.
>
> **Any advantages?**
>
> It's hard to compare our method fairly with the first type of methods due to the difference in compression philosophy. While they seem to be better in terms of compression ratio and task performance, they are impractical in real-world scenarios because they cannot use FlashAttention and require re-computing the relevance to the user prompt for every inference. Compared to the second type (e.g. ToME), our method is more refined because it *converts an image to natural language* and merges similar tokens directly via text strings. Thus it has a lower compression ratio than ToME, but better performance on downstream tasks.
>
> A comparison with other works is highly necessary, and we'll add this part to our paper!
>
> **Training-free RoPE scaling experiment**
>
> We first filtered samples related to spatial reasoning from the GQA training set based on the keywords related to directions, resulting in $4.5k$ samples. Based on this subset, we adjusted $\alpha$ and $p$ to achieve better performance. We used $p=6$ and $\alpha=49$ to test on What’s Up A/B, COCO/GQA-Spatial1/2, which yielded the results in Table 2.
>
> As for VSR, because its QA format is judging correctness (true/false) rather than providing a direct answer, we tuned its parameters separately. We performed a grid search on the VSR validation set (random split). We select $p=8$ and $\alpha=149$ to test on the VSR test set. We hope our explanation clarifies your doubts. We'll add these details to the paper.
>
> **Questions**
>
> **(1) About DeepStack**
>
> We believe it is possible, but there is some points to note.
> On the one hand, we believe the sampling method for DeepStack must be considered. We think the 2D Spatial sampling method is more suitable for our analysis, because the resulting low-resolution images could preserve the structure of the original image.
>
> On the other hand, we think the size of the objects also matters. For larger objects (i.e., those containing a sufficient number of tokens), we believe our theory can be applied by analyzing each low-resolution image individually. However, for smaller objects (e.g., those containing only one or two visual tokens), there is a risk that the object might be missing in certain low-resolution images, which would affect our analysis.
>
> **(2) About 4D RoPE（or 3D?）**
>
> The concept of the 'when' pathway is very interesting! The relationship between frames in video data involves an important question: Does the VLM possess a capacity similar to predictive coding in the human brain? That is, when performing video understanding, does the VLM utilize prior knowledge to predict the state of future frames, even if they are absent? For example, given a frame of a ball in the air, can the VLM know that in the next frame, the ball should be moving downwards? And why? We believe it's possible, but it still requires investigation from both theoretical and experimental angles.
>
> For 3D RoPE, 1/3 of the model dimensions represents time. Theoretically, we can separate the temporal part via an analysis similar to that in Section 4.2 and compare the representations at different time steps. Empirically, one possible approach would be to play the video in reverse and then observe what changes occur inside the model (it's just an idea we find interesting and might be worth trying).
> ___
> Thanks again for your time devoted in our work. Hope this response could further relieve your concerns!
>
> **Reference**
>
> [1] Token Merging: Your Vit But Faster https://arxiv.org/pdf/2210.09461

---

> > ### Comment · Reviewer_fdDb · 2025-11-27
> >
> > __It's hard to compare our method fairly with the first type of methods__
> >
> > I don't see a reason that flash-attention can not be used, as pooling can be applied before it. I think that a numeric comparison will be extremely useful here.
> >
> > __A comparison with other works is highly necessary__
> >
> > I agree, and I can only encourage the authors to add those.
> >
> > For now, and after reading the comments of other reviewers, I will keep the score as is.

---

> > > ### Author Response · Authors · 2025-11-27
> > >
> > > Thank you for your response!
> > >
> > > **Why most of the token pruning methods (based on instruction relevance) cannot use FlashAttention ?**
> > >
> > > The main reason is that these methods need to get the full attention scores in order to measure the relevance between visual tokens and the user prompt. They prune the visual tokens that have a low relevance to the user prompt. However, the core optimization strategy of FlashAttention is to avoid materializing the massive $N \times N$ attention matrix (where $N$ is the sequence length) in the GPU's main memory (HBM/DRAM). Because the computation is fused and tiled, the full $N \times N$ attention score matrix never exists as a single, saved tensor in memory at any point. It's computed block-by-block, and the intermediate scores are immediately used to update the final output and then discarded from SRAM.
> > >
> > > Take *Fast-V* as an example, it first computes the attention received from all other tokens for each visual token, and prune $(1-r)|V|$ visual tokens in the input of a layer, where $r$ is the compression ratio, and $|V|$ is the number of visual tokens. Therefore the input length in the next layer would be reduced. However, Fast-V requires the full attention map to compute the attention scores, which is unavailable when using FlashAttention. As far as we know, most of the token compression methods  require the full attention maps for the computation of the relevance to the user prompt.
> > >
> > > While our method as well as *ToMe* perform don't require the full attention maps. Our method performs token merging before the visual sequence enters the LLM, and requires only one calculation, while the instruction-relevance based methods often need to perform token pruning in multiple layers. As for ToMe, it performs token merging based on the attention keys, just before the calculation of attention maps.
> > >
> > > As for the quantitative analysis, we have compared our method with ToMe on VQA-v2, GQA, SQA, TextVQA, MMBench, and POPE. The results are shown in our response to *Reviewer ywbf*, which demonstrate that our method could capture the precise semantics of visual tokens better than ToMe. Hope this response could further clarify your doubts. Thanks again for your time!

---

### Official Review · Reviewer_RfD1 · 2025-10-31

**Soundness:** 3
**Presentation:** 3
**Contribution:** 3
**Rating:** 6
**Confidence:** 3

**Summary:**

This paper investigates how Vision-Language Models (VLMs) process visual information by decomposing their mechanisms into object recognition and spatial perception, inspired by the dual-stream hypothesis of human vision. The study reveals that VLMs recognize image content in two stages—progressing from low-level attribute detection to high-level semantic understanding—and uncovers the geometric structure underlying their positional  representation. Building on these insights, the authors propose a token compression algorithm and a RoPE scaling technique to enhance decoding efficiency and spatial reasoning. Overall, the work deepens understanding of VLM internals and offers principled guidance for future model design.

**Strengths:**

* This paper conducted comprehensive theoretical and experimental analyses.
* The paper connects theoretical analysis with applications. Based on findings on the visual processing and spatial perception characteristics of VLMs, this paper introduces an instruction-agnostic token compression method, which reduces image sequence length during decoding, and RoPE scaling, which enhances the spatial reasoning capabilities of VLMs.

**Weaknesses:**

* The paper does not quantitatively compare the proposed token compression method with other existing compression techniques, such as [1] and [2], in terms of inference efficiency and training cost. The proposed method also requires additional training of the visual decoder, which may limit its application scenarios.
* In Table 1, methods 2 and 3 exhibit a more significant performance decline on TextVQA than on other tasks. What accounts for this difference?
* There are several instances in the paper where LaTeX quotation marks are incorrectly formatted. For example, ”In which direction is A relative to B?” in line 316.

[1] Lin, Zhihang, et al. "Boosting multimodal large language models with visual tokens withdrawal for rapid inference." Proceedings of the AAAI Conference on Artificial Intelligence. Vol. 39. No. 5. 2025.

[2] Zhu, Yuke, et al. "Focusllava: A coarse-to-fine approach for efficient and effective visual token compression." arXiv preprint arXiv:2411.14228 (2024).

**Questions:**

Please refer to the Weaknesses part.

---

> ### Author Response · Authors · 2025-11-19
>
> Thanks a lot for your time on our work. I'll try my best to answer your concerns!
>
> **(1) Comparison with other methods**
>
> Thank you for pointing this out! We believe a comparison with other token compression methods is necessary. Due to page limitations, we had to prioritize the most crucial parts for inclusion in the paper. Here, we would like to briefly compare our method with prior work.
>
> **Inference Efficiency**
>
> First, we provide a theoretical derivation for the compression ratio of our method:
>
> Since we only add an extra visual decoder after the modality connector compared to the original model, we ignore the computational cost of the ViT and the modality connector, and only consider the LLM. Furthermore, we focus only on the main overhead—the matrix multiplications within the Attention and MLP modules—while ignoring the text embedding, Softmax, activation functions, and the final language head.
>
> Assume the LLM has $L$ layers, the model dimension is $d$, and the intermediate dimension of the MLP is $d'$. The visual decoder $\varphi$ has a size of $\mathbb{R}^{d\times|\mathcal{V}|}$, where $\mathcal{V}$ is the vocabulary. We consider the input of a single image, where the visual sequence length is $n$, and we ignore the length of the instruction. The computational cost of the original model is:$$C = (8nd^{2}+4n^{2}d+4ndd’)L \quad (\text{FLOPs})$$When the compression ratio is $r$, the computational cost of our method is:$$C’=[8n(1-r)d^{2}+4n^{2}(1-r)^{2}d+4n(1-r)dd’]L+2nd|\mathcal{V}| \quad (\text{FLOPs})$$Therefore, we can measure inference efficiency by $\frac{C’}{C}$. For LLaVA-1.5-7B, the compression ratio $r$ of our method can reach $0.58$, corresponding to $\frac{C’}{C}=43.82$%, which is highly efficient. We find the papers you recommended to be excellent. In fact, we also briefly discussed prior work in the appendix. Prior methods are mainly based on instruction relevance, or the similarity between visual tokens. The first type, such as Fast-V, SparseVLM, VisPruner, FocusLLaVA, VTW, etc., perform compression via the relevance of visual tokens to the instruction, whereas our method solely examines visual redundancy within the image. Due to this difference in compression philosophy, it is difficult to compare our method fairly with them. While they seem to have an advantage in terms of compression ratio and downstream task performance, they are impractical in real-world scenarios (e.g. chatbots involving long text and multi-turn conversations) because they cannot use FlashAttention and require re-computing the relevance to the user prompt for every inference.
>
> Compared to the second type, represented by methods like ToMe, our method is more refined because it finds similar visual tokens by *converting an image to texts* and merges the tokens using the interpretable text token strings. Therefore our method has a lower compression ratio than ToME, but significantly better performance on downstream tasks.
>
> Thank you for sharing the references with us. We will include both qualitative and quantitative analysis of these works in the final version of the paper.
>
> **Training cost of the visual decoder**
>
> This is a limitation of our work compared to training-free methods. While training such a visual decoder is quite simple. As detailed in Appendix G, it takes approximately 5 hours on an A40 GPU and is highly data efficient. For LLaVA-1.5-7B, its visual decoder only requires training for $6k$ steps with a batch size of $16$, using less than $100k$ of unlabeled data. Therefore, we believe this training cost is acceptable.
>
> **(2) Performance on TextVQA**
>
> We are glad to discuss this point with you. The reason is actually quite simple: the proportion of images containing text in the data used to train the visual decoder is low. The training data for the visual decoder consists of two parts: $148k$ images from the GQA dataset and $22k$ images from the TextVQA training set. All images in TextVQA contain text. As you can see, the proportion of text-containing images, which require OCR capability, is relatively low. Furthermore, the area occupied by text within a single image is often small. This results in fewer text-related supervisory signals during training, and the OCR capability of the visual decoder was not fully developed.
>
> Since we only used the token compression experiment in Section 5.1 as an application example, we did not spend much effort on enhancing its performance. However, simple preliminary verification has already demonstrated the effectiveness of the method. Therefore, we believe that we can completely improve the results on the TextVQA test set by increasing the proportion of text-containing images in the training data. In summary, we hope for your understanding on this point.
>
> **(3) Typos**
>
> We’ll fix them up!
> ___
> Thanks again for your time devoted in our work. Hope this response could further relieve your concerns!

---

> > ### Comment · Reviewer_RfD1 · 2025-11-28
> >
> > Thank you for your reply! I have no other questions and I will keep my rate of the paper.

---

### Official Review · Reviewer_ywbf · 2025-10-31

**Soundness:** 3
**Presentation:** 3
**Contribution:** 2
**Rating:** 4
**Confidence:** 3

**Summary:**

This paper provides a deep interpretability analysis of Vision-Language Models (VLMs) through the lens of the brain’s dual-stream hypothesis, examining how models achieve both object recognition (“what”) and spatial perception (“where”). For the “what” pathway, logit lens decoding reveals a two-stage process in token representations: early detection of low-level features and later semantic refinement. For the “where” pathway, theoretical and empirical investigations of 2D RoPE demonstrate that spatial relations are geometrically encoded in attention maps. Leveraging these findings, the authors introduce (1) an instruction-agnostic token compression algorithm using run-length encoding, and (2) RoPE scaling to enhance positional signals and spatial reasoning. Experiments on LLaVA and Qwen2.5-VL validate both the insights and practical effectiveness of the proposed methods.

**Strengths:**

1. The paper offers a systematic and mechanistic analysis of visual processing in VLMs. The application of the logit lens to visual tokens is particularly effective, converting otherwise opaque embeddings into interpretable token maps and enabling fine-grained examination of object recognition dynamics.

2. The paper provides a rigorous theoretical analysis of how 2D RoPE encodes spatial relationships, with empirical validation via PCA, object erasure, and intervention experiments. The findings on collinearity of “left/right” and orthogonality of “left/behind” directions convincingly illustrate the geometric structure in the learned representations.

3. The paper features extensive ablation studies, comprehensive training protocols, and thoughtful handling of edge cases, such as top-1 versus top-2 token filtering for compression. Evaluation across multiple models (LLaVA, Qwen2.5-VL, InternVL) and datasets (GQA, POPE, What’s Up) further enhances the generalizability of the findings.

**Weaknesses:**

1. The study's focus on four basic directional relationships (“left,” “right,” “front,” “behind”) excludes more complex spatial configurations (e.g., “top-left,” “surrounding,” “partially occluded”), thereby limiting the scope of the “where” pathway analysis.

2. The logit lens approach assumes that visual representations can be linearly decoded into semantic tokens using the language model’s unembedding matrix. However, this assumption may not hold for earlier ViT layers, whose features are not yet aligned with the final modality connector, as noted in Appendix A. Employing per-layer projection heads could offer a more robust alternative.

3. Although the compression method is instruction-agnostic, it necessitates training a visual decoder through knowledge distillation, which introduces additional engineering overhead. Furthermore, the assertion that "random selection outperforms mean pooling" is made without theoretical justification, indicating a need for more thorough investigation in future work.

**Questions:**

See weaknesses.

---

> ### Author Response · Authors · 2025-11-19
>
> Thanks a lot for your kind suggestions. We'll try our best to answer your concerns!
>
> **(1) Analysis on more complex spatial relationships**
>
> This is just what we considered when writing this paper, and we believe that the current theoretical and empirical analysis supports more complex spatial relationships. Taking 'left-front' as an example. According to Section 4.2, the coordinates of the satellite object should be $(-\frac{\sqrt{2}}{2}m, \frac{\sqrt{2}}{2}m)$. Its representation is therefore:
>
> $$ h_{A}^{left-front} = Re[q_{A}^{X}{k_{A}^{X}}^{\*} + q_{A}^{Y}{k_{A}^{Y}}^{\*}]v_{A} + Re[q_{A}^{X}{k_{B}^{X}}^{\*}e^{i(-\frac{\sqrt{2}}{2} m\theta)} + q_{A}^{Y}{k_{B}^{Y}}^{\*}e^{i(\frac{\sqrt{2}}{2} m\theta)}]v_{B} $$
>
>
> It can be seen that compared to Equation (24) and (30) in Appendix F.1, the coefficient of $v_{B}$ in the equation above is equivalent to the fusion of $h_{A}^{left}$ and $h_{A}^{front}$. Therefore, the spatial-level semantics simultaneously contain components of both 'left' and 'front.' Given that the four basic directions in Figure 7(a) present a very standard spatial geometry, we believe that more complex directions like 'left-front,' which is a linear combination of 'left' and 'front' vectors, will exhibit the expected geometric structure.
>
> Due to page limitations, we only discussed the four basic spatial relationships. We'll consider adding a detailed discussion on more complex spatial relationships you mentioned in the final version of the paper. Thank you for pointing this out!
>
> **(2) Per-layer projection heads**
>
> Yes! from a more rigorous perspective, per-layer projection heads are more reasonable. While we believe the modality connector is capable of handling the outputs of the earlier layers of the ViT, at least reasonable from the perspective of Transformer mechanisms:
> * According to the theoretical analysis by Elhage et al. [1], with the existence of the *residual stream*, every layer in a Transformer contains the output of all previous layers. This means that *the last layer of a ViT is essentially a linear superposition of the current layer's output and the outputs of all previous layers*.
> * During VLM pre-training, on a macro level, the output from the last layer of the ViT is aligned with the language space via the modality connector. While from a micro perspective, the modality connector actually processes the last layer of the ViT, along with the outputs of all preceding layers. Thus it should have developed the ability to handle early ViT layers.
>
> On the other hand, training per-layer projection heads would mean that whenever we get a new model, we need to train a new projection head for every layer of it.  While we expect to contribute a fast and effective analysis tool to the research community that is ready-to-use, i.e., quick analysis results can be obtained for any layer of any model.
> Therefore, we would like to conduct a dedicated study on this rigorous approach in the future.
>
> **(3) The token compression method**
>
> **The visual decoder involves extra training**
>
> We confess that this is a limitation of our work compared to training-free methods. However, we would like to point out that training such a visual decoder is not as time-consuming as one might imagine. As detailed in Appendix G, training the visual decoder only takes approximately 5 hours on an A40 GPU and is highly data efficient. For LLaVA-1.5-7B, its visual decoder only requires training for $6k$ steps with a batch size of $16$, using less than $100k$ of unlabeled data.
> Therefore, we believe this training cost is acceptable.
>
> **Random selection vs. mean pooling**
>
> Thank you for advice~ It’s hard to rigorously prove which is theoretically better, but we are willing to share our perspective here: Assume the visual embeddings are $V=(v_1, v_2, v_3, v_4, v_5) \in \mathbb{R}^{5 \times D}$, where $D$ is the dimension of the language model. If the text tokens corresponding to $V$, decoded by the visual decoder $\varphi$, are $(\text{"banana"}, \text{"dog"}, \text{"dog"}, \text{"dog"}, \text{"monkey"})$, then the mean pooling method compresses the visual embeddings into $(v_1, (v_2+v_3+v_4)/3, v_5) \in \mathbb{R}^{3 \times D}$, where the embedding of "dog" is the average of $v_2$, $v_3$, and $v_4$.
>
> When using random selection, we have a $1/3$ probability of selecting $v_2$ or $v_4$. As they are adjacent to "banana" or "monkey", their embeddings will be more influenced by the semantics of the adjacent objects. Mean pooling, on the other hand, can emphasize the semantics of "dog" by introducing $v_3$, since $v_3$ gets less semantic interference from other objects.
> We hope this clarifies our explanation, and we'll add discussions about this issue in our paper.
>
> ___
> Thanks again for your time, and we'll appreciate it if you could re-evaluate our work. Looking forward to your reply !!
>
> **Reference**
>
> [1] A Mathematical Framework for Transformer Circuits https://transformer-circuits.pub/2021/framework/index.html

---

> > ### Comment · Reviewer_ywbf · 2025-11-27
> >
> > Thank you for the author's response.
> > My questions have been largely addressed.
> >
> > Overall, in my personal view, this paper provides a thorough analysis and proposes corresponding optimization methods based on the analytical findings. However, regarding the proposed methods:
> >
> > (1) Token Compression: While the authors also analyze and discuss ToMe, both approaches are essentially based on compressing visual tokens using semantic information. The authors claim their method performs better than ToMe, but there appears to be no experimental table to support this claim.
> >
> > (2) RoPE Scaling: The authors' approach is similar to the linear RoPE handling in transformers [1]. However, transformers already provides a richer set of RoPE optimization methods.
> >
> > Therefore, overall, the contribution of the methodology section seems somewhat weak.
> >
> > [1] https://github.com/huggingface/transformers/blob/main/docs/source/en/internal/rope_utils.md

---

> > > ### Author Response · Authors · 2025-11-27
> > >
> > > We are glad that we were able to address the initial question you raised.
> > >
> > > Regarding the two new questions you've posed, we would like to offer a more detailed discussion to minimize any misunderstandings and clarify concerns.
> > >
> > > **Question 1: Comparison with ToMe**
> > >
> > > To ensure the rigor of this work, we have conducted further experiments to compare the performance of ToMe and our method.
> > > ToMe was originally developed for token pruning on a single ViT and was not tested on VLMs. Therefore, we reproduced the method from the ToMe paper on LLaVA-1.5-7B and Qwen2.5-VL-7B. ToMe divides the visual token sequence into two sets of equal size and uses a method called "bipartite soft matching," which merges the top-$r$ groups of most similar tokens, and then merges the two sets back together. The hyperparameter that needs to be considered in this process is the reduction rate $r$. Following the settings in the ToMe paper, we selected a *constant schedule* (where $r$ remains unchanged in all ViT layers).
> > > Furthermore, since our token compression includes three distinct methods corresponding to different reduction rates, we set the compression rate $r$ for ToMe to match these rates to ensure fairness in the experimental comparison.
> > >
> > > We evaluated the performance on VQA-v2, GQA, SQA, TextVQA, MMBench, and POPE. For each benchmark of each method, we run 5 times with different random seeds, and the results are the average of the 5 runs. The evaluation results are as follows:
> > >
> > > | Methods & Datasets | VQA-v2 | GQA | SQA | TextVQA | MMBench(EN) | POPE | Reduction rate (%) |
> > > | :--- | :---: | :---: | :---: | :---: | :---: | :---: | :---: |
> > > | **Original Decoding (Upper Bound)** | | | | | | | |
> > > | LLaVA-1.5-7B | 48.93 | 60.50 | 48.46 | 45.01 | 53.73 | 85.96 | / |
> > > | Qwen2.5-VL-7B | 79.04 | 61.20 | 80.55 | 78.21 | 84.68 | 89.23 | / |
> > > | **M1: Runlength compression** | | | | | | | |
> > > | LLaVA-1.5-7B + *ToMe* | 45.30 | 58.72 | 45.51 | 39.44 | 50.02 | 84.21 | 27.83 |
> > > | LLaVA-1.5-7B + *our method* | **49.24** | **61.32** | **47.37** | **43.16** | **51.40** | **86.00** | 27.83 |
> > > | Qwen2.5-VL-7B + *ToMe* | 74.83 | 58.64 | 77.52 | 74.66 | 83.92 | 86.23 | 16.19 |
> > > | Qwen2.5-VL-7B + *our method* | **77.91** | **60.80** | **80.71** | **76.33** | **84.27** | **87.46** | 16.19 |
> > > **M2: Runlength compression + Remove punctuation (Look at Top1 token)**
> > > | LLaVA-1.5-7B + *ToMe* | 45.33 | 58.27 | 44.24 | 34.95 | 45.23 | 83.39 | 58.35 |
> > > | LLaVA-1.5-7B + *our method* | **48.59** | **60.14** | **46.59** | **35.04** | **47.33** | **85.66** | 58.35 |
> > > | Qwen2.5-VL-7B + *ToMe* | 76.15 | 56.70 | 77.38 | 68.43 | 80.26 | 84.41 | 45.00 |
> > > | Qwen2.5-VL-7B + *our method* | **77.24** | **58.69** | **79.70** | **71.80** | **83.39** | **85.70** | 45.00 |
> > > | **M3: Runlength compression + Remove punctuation (Look at Top2 tokens)** | | | | | | | |
> > > | LLaVA-1.5-7B + *ToMe* | 47.70 | 60.20 | 44.52 | 34.12 | 48.90 | 84.41 | 48.55 |
> > > | LLaVA-1.5-7B + *our method* | **49.12** | **61.06** | **47.18** | **38.53** | **50.45** | **85.90** | 48.55 |
> > > | Qwen2.5-VL-7B + *ToMe* | 75.20 | 58.33 | 78.21 | 72.44 | 81.67 | 85.66 | 32.09 |
> > > | Qwen2.5-VL-7B + *our method* | **77.38** | **60.03** | **80.48** | **74.78** | **83.58** | **86.83** | 32.09 |
> > >
> > > As the results show, our method achieves superior performance across multiple benchmarks at the same reduction rate. As you mentioned, both ToMe and our method perform token merging from a semantic perspective. However, the key difference is that our method directly decodes the visual tokens into *human-interpretable, natural language text tokens* from the view of logit lens, whereas ToMe merely replaces the cosine similarity with the similarity between the attention keys. This latter approach lacks good interpretability, and its effectiveness may vary with changes in model architecture, leading to less stable performance compared to our method. Therefore, our method holds a greater advantage. We hope that our response would further help you understand our method.
> > >
> > > We will include these quantitative analysis results in our paper.

---

> > > > ### Author Response · Authors · 2025-11-27
> > > > **Continue**
> > > >
> > > > This might be a misunderstanding~ Our RoPE scaling and the linear RoPE scaling in the *transformer* library you mentioned refer to two completely **different techniques**.
> > > >
> > > > **The linear RoPE in transformers aims to handle long-context scenarios**
> > > >
> > > > The linear RoPE you mentioned is a linear interpolation method used to extend the context length. For example, the maximum text length a model encounters during pre-training is $L_{train}$, but during inference, the user's input text length $L_{new}$ might be greater than $L_{train}$. To adapt the length to the range that RoPE can handle, we can scale $L_{new}$ down to a range smaller than $L_{train}$. That is, $s = \frac{L_{new}}{L_{train}}$, where $s$ is the scaling factor. During the inference phase, we multiply the original position IDs by this factor $s$ to achieve position scaling. In this case, the trigonometric frequency in the RoPE formula becomes $m'\theta = \frac{m}{s}\theta = \frac{m}{L_{new}} L_{train}\theta$, where $m$ is the position ID.
> > > >
> > > > **The RoPE scaling in our paper aims to enhance the spatial reasoning capability of VLMs**
> > > >
> > > > The RoPE scaling we propose is essentially an adaptive scaling of the RoPE frequencies corresponding to different dimensions in the model. The motivation for our method comes from observing the phenomenon of dimensional decay in RoPE. As discussed in Section 5.2, for a certain RoPE rotation angle $\theta_i$ (where $i$ is the index of the dimension group), as $i$ increases (i.e., as the model dimension goes from low-dimension to high-dimension), the rotation angle $\theta_i$ exhibits an extreme decay, as shown in Figure 24(a) (Appendix H). In this situation, the high dimensions in the model cannot effectively represent positional information. This is because when $\theta_i$ is small, the angular frequency of RoPE becomes very small, meaning RoPE becomes highly insensitive to changes in distance. At this point, only very large changes in distance can cause a change in the magnitude of the model's representation. This is extremely detrimental for a VLM when judging spatial relationships.
> > > >
> > > > Therefore, we believe it is necessary to adaptively compensate for distance in the high-dimensional part of the model to make up for the loss of positional information caused by frequency decay. As shown in Figure 24(b) in Appendix H, we amplify the position IDs in the high-dimensional part by multiplying them with a scaling factor $g(i) = [1 + \alpha(\frac{2i}{d})^p]$ (which is greater than 1), where $d$ is the total dimension of the model, $i$ is the index of the dimension group, $\alpha$ is used to adjust the scaling strength, and $p$ determines which dimensional regions receive stronger compensation. In this case, the RoPE frequency changes to $m[1 + \alpha(\frac{2i}{d})^p]\theta_i$.
> > > >
> > > > **The difference between two methods: different tasks, different directions**
> > > >
> > > > Based on the above analysis, you may realize that our method and linear RoPE are actually moving *in opposite directions*. Our method adaptively **amplifies** the RoPE frequency in the high-dimensional areas, while linear RoPE **shrinks** all RoPE frequencies. This is entirely due to the different problems they aim to solve: our method addresses the VLM's insensitivity to positional information in the high-dimensional region, while linear RoPE addresses the problem of context length extrapolation for LLMs in long-text scenarios. Therefore, our method belongs to a different domain from the one you mentioned.
> > > >
> > > > A worthy question is whether our method leads to a degradation of general capabilities. Regarding this, as stated in the experimental analysis in Section 5.2 of the main text, we tested the performance of the RoPE scaling method on general capabilities using MMBench. The experimental results show that our method can even *improve* the VLM's general capabilities (acc 85.52% vs. 85.45%), which proves that our method is highly robust, capable of enhancing spatial abilities while maintaining general capabilities without loss. As for context capability, since ViT only processes a single image, even if we input a high-resolution image (e.g., $1920 \times 1080$), the visual tokens sequence length after patchifying is only 2645, which is far less than the context length involved in LLM long-text scenarios (often 100K or even 100M). Thus, our method does not need to consider long-text scenarios.
> > > >
> > > > ___
> > > > In summary, we hope our reply has clarified your doubts, and we thank you again for your valuable suggestions.
> > > >
> > > > We would be very grateful if you could reconsider and accept our work. Looking forward to your reply!

---

### Official Review · Reviewer_fYWu · 2025-11-01

**Soundness:** 3
**Presentation:** 3
**Contribution:** 4
**Rating:** 8
**Confidence:** 3

**Summary:**

First, they investigate _how_ the vision encoder of the VLM encodes what an object is, and they do so by skipping ViT layers and using logit lens on the LLM to see what the LLM processes. They show that early layers encode high-level attributes and later layers encode specific repesentations of the objects. Then, they study at how VLMs perceive spatial relation through a theoretical and empirical analysis. Lastly, they propose two applications based on their findings: token compression and enhancing spatial reasoning.

**Strengths:**

* They use logit lens on the LLM to study how the Vision Encoder works: this is a particularly insightful idea that I had not previously considered. And it seems to work extremely well, especially with further validation from POPE polling.
* It is timely to study how VLMs do spatial perception as they are becoming better at it, and the paper does a sound, comprehensive study of it. While the theoretical study is limited to only two objects, I find this to be a necessary constraint.

**Weaknesses:**

* The segmentation map keywords seem to be chosen by the authors. This may mean that the results could be due to human-confirmation bias here, if the authors first saw the logit lens results then came up with the keywords.
* The applications and results don't seem very strong, but nevertheless serve as further empirical validation of their findings.

**Questions:**

* What was the method used to come up with the keyword set? If it was what I mentioned, is there some more principled way to do it?
* For the token compression application, the tokens embeddings only become text-like in the later layers of a LM. I assume you are compressing the tokens at the input to the LM (i.e. replacing or truncating the visual embeddings.) How are you compressing the tokens, or what are you replacing the visual embeddings with? I understand it to be text tokens from the logit lens. If so, have you tried merely replacing all visual embeddings with logit lens tokens? (I'm not sure if this is what "original decoding" refers to, or whether original decoding is referring to merely using the original visual embeddings)

---

> ### Author Response · Authors · 2025-11-19
>
> Thanks a lot for your kind suggestions on our work. We are happy to know that you are interested in our work!
>
> We'll try our best to answer your concerns.
>
> **(1) Ways to get the keywords set for the segmentation map (Weakness 1 & Question 1)**
>
> When drawing the segmentation map, we adopted a pipeline involving automated labeling followed by manual verification. Specifically, the construction of the keyword set follows two steps:
>
> * Using LLM to generate keywords for each primary object. The GQA dataset labels the name of each primary object in every image, and these words can be used directly. However, the token map we refer to when drawing the segmentation map contains a variety of tokens. The most common case is that a single object may have multiple names (e.g., 'child,' 'kid,' 'boy' can all refer to a person in the image). Therefore, in the first stage, we use an LLM (Qwen2.5-7B) to generate multiple synonyms for each object name. This step is automated.
>
> * Human review. As you mentioned, we do supplement the keyword set from the LLM based on the token map. The motivation for doing this is not only to better present the segmentation map, but to faithfully show the VLM's representation. In this stage, we primarily address the multi-lingual issue (i.e., tokens for an object might belong to languages other than English) and tokens that are not synonyms but still refer to the same object. For instance, regarding the water surface in Figure 11 (a), 'water' and 'reflection' may not seem like the same concept, but both can represent the same primary object. Such cases are difficult for an LLM to handle well, thus requiring human modification.
>
> Therefore, we can assure that the keyword set is faithful to the original image to the greatest extent possible, rather than being directly compiled from the token map. We will include the LLM prompt in the appendix of the final version of this paper. Thank you for pointing this out!
>
> **(2) Further explanations of the token compression method (Question 2)**
>
> To explain it more intuitively, let us consider a simple example: Assume the visual embeddings are $V=(v_1, v_2, v_3, v_4, v_5) \in \mathbb{R}^{5 \times D}$, where $D$ is the dimension of the language model. If the text tokens corresponding to $V$, decoded by our trained visual decoder $\varphi$, are (“dog”, “dog”, “dog”,“banana”, “ground”), then, based on the idea of Run-Length Encoding (RLE) as described in Algorithm 2, this text sequence is compressed into: (“dog”, “banana”, “ground”). Since what we need to compress is the image sequence, we can choose the mean pooling method, transforming the input sequence into $((v_1+v_2+v_3)/3, v_4, v_5) \in \mathbb{R}^{3 \times D}$. This is the method we used in our experiments.
>
> Furthermore, original decoding refers to the inference of the VLM without any intervention, i.e., the baseline without using any token compression method.
>
> “Replacing all visual embeddings with logit lens tokens” would likely not work in our view, because the visual embeddings have been processed by the ViT, involving multiple layers of attention and MLP. Simply replacing the visual embeddings with text tokens based on the decoded text embeddings would lead to a loss of the semantic-level spatial structure between the image patches, causing them to degenerate into isolated image tokens.
>
> ---
> Finally, we hope our response could further relieve your concerns. Thank you again for your time on our work !!!

---

### Author Response · Authors · 2025-11-29

**Summary of the Discussion**

We would like to thank all the reviewers and the AC for the time and effort on our paper. We are pleased to provide a summary of the review process and the ongoing discussions.

Firstly, the initial average score for our work was 6.0, which remains unchanged at this time. Our discussions with the four reviewers are summarized as follows:
* **Reviewer fYWu (Score: 8):** We provided a detailed account of the keyword selection process for the segmentation map and clarified the specific methodology of our token compression algorithm. The reviewer has not yet replied, but we believe we have addressed all their questions to the best of our ability.
* **Reviewer ywbf (Score: 4):** We provided detailed explanations addressing the three weaknesses raised by the reviewer. The reviewer acknowledged this by stating *"My questions have been largely addressed"*. However, they subsequently raised two new questions derived from the comments of other reviewers. For question (1), we provided further detailed experimental results to support our conclusions. For question (2), we believe this stemmed from a misunderstanding and provided a detailed analysis to clarify the difference between our method and the one the reviewer mentioned. We have not yet received a response from the reviewer concerning these latest explanations.
* **Reviewer RfD1 (Score: 6):** We successfully addressed the reviewer's questions. The reviewer thanked us for the clarification and raised no further issues.
* **Reviewer fdDb (Score: 6):** Our response resolved the majority of the reviewer’s concerns. Nevertheless, the reviewer appeared to remain confused about the point that "most token compression methods are incompatible with FlashAttention." We provided a further explanation regarding this point but have not yet received a reply.

In conclusion, we are confident that we have successfully resolved the main issues raised by all four reviewers. Based on this review process, we summarize the contributions of our work as follows:
1. We categorize VLM capabilities into object recognition (the “what” way) and spatial perception (the “where” way). We conduct in-depth analysis of these capabilities using a combination of theoretical and empirical analyses, offering a novel perspective for VLM interpretability research.
2. Based on the results of our interpretability analysis, we propose a token compression algorithm and a technique to enhance the spatial reasoning capability of VLMs, demonstrating the practical application and value of our findings.

We thank all the reviewers and the AC for their valuable time and effort during this review process.

---

### Meta-Review · Area_Chair_QMr8 · 2026-01-05

**Summary:**

Across reviewers, the overall evaluation is moderately positive (8/4/6/6).

One reviewer (fyWu) is strongly supportive of acceptance (score 8). Two reviewers (RfD1 and fdDb) provide borderline-positive evaluations (score 6), acknowledging the paper’s analytical depth while remaining neutral regarding the final accept/reject decision. One reviewer (ywbf) is borderline-negative (score 4).

**Reviewer Concerns:**

- Reviewer fyWu raised concerns about potential bias in segmentation keyword selection and the limited empirical impact of the proposed applications; I think these issues were addressed in the rebuttal.

- Reviewer ywbf questioned the scope of the spatial analysis, the robustness of logit-lens decoding for early layers, and the complexity of the token compression method; while largely addressed, the reviewer still views the methodological contribution as somewhat weak. I agree with this reviewer.

- Reviewer RfD1 raised concerns regarding comparisons with prior token compression methods, training overhead, performance degradation on TextVQA, and minor presentation issues; these concerns are mostly resolved after rebuttal.

- Reviewer fdDb noted issues with paper organization, insufficient comparison to prior work, and potential data leakage in training-free RoPE scaling; the authors clarified these points,  I also think that additional comparisons would further strengthen the paper.

**Reviewer Scores:**

8/4/6/6

The paper is consistently rated as sound and well-presented, with no reviewer raising fundamental concerns about correctness or significance.  Lower scores primarily reflect scope limitations, methodological strictness, and experimental completeness.

In conclusion, the paper provides meaningful mechanistic analysis of VLMs, makes effective use of logit-lens techniques, and offers a careful theoretical and empirical study of 2D RoPE. At the same time,  I note that the work would be strengthened by broader spatial analysis and more extensive quantitative comparison with prior token compression methods.

---

### Decision · Program_Chairs · 2026-01-26

Accept (Poster)